# Saint: Spatial Guidance for Inpainting

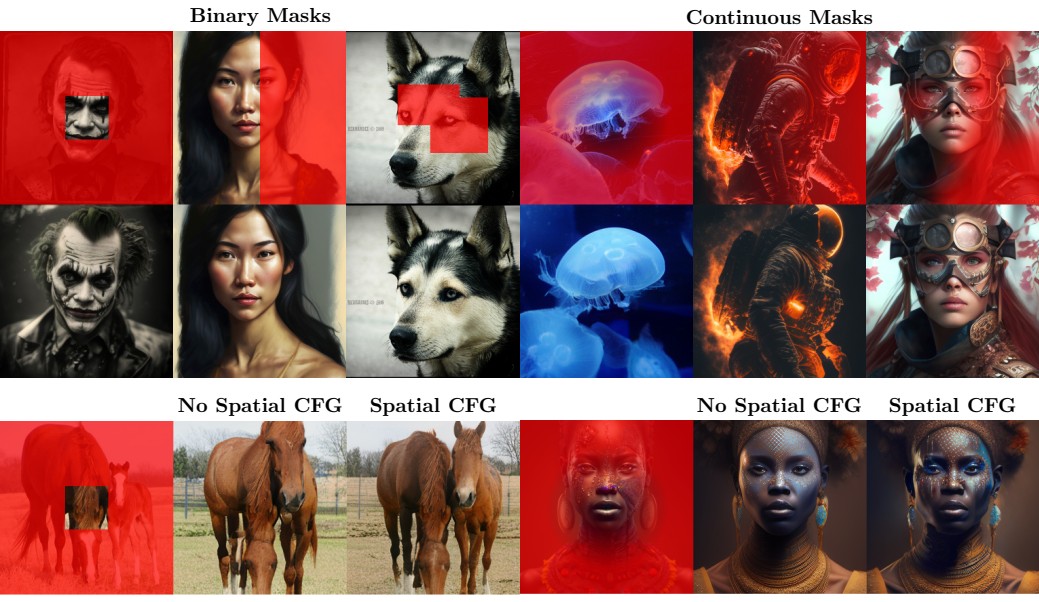

Figure 1: We present **Saint**, a method that fine-tunes large-scale latent Diffusion Transformers as Spatial Reasoning Models tailored for image inpainting. By performing masking via noising, it achieves high-quality inpainting for binary as well as continuous masks (shown in red). We further introduce Spatial Classifier-Free Guidance to boost fidelity and alignment with the input image.

## Abstract

We introduce Saint, a framework for image inpainting with large-scale diffusion and flow-based transformers in a latent multi-variable setup. Existing methods for latent image inpainting rely on RePaint-like sampling or mask concatenation, which either does not make use of the masked image as strong conditioning at all or neglects the fact that the denoising model has been already trained for masking via noising. In contrast, Saint fine-tunes pre-trained Diffusion Transformers (DiTs) as Spatial Reasoning Models (SRMs) with varying noise levels across masked and unmasked regions, allowing to condition the model directly via the partially noised latent. This more effective conditioning scheme improves inpainting performance on binary masks and further extends to continuous masks. Moreover, the multi-variable formulation of SRMs allows us to formulate a Spatial Classifier-Free Guidance strategy tailored for inpainting as well as a token-caching scheme for efficient local edits. We evaluate Saint on ImageNet1k and JourneyDB datasets for a variety of inpainting scenarios and show that it consistently improves on the state of the art in generative and reconstruction metrics. Our codebase and the models will be released publicly upon acceptance of the paper.

## 1 Introduction

Inpainting masked regions in images is a challenging and relevant task that requires strong prior knowledge about complex image distributions. Given the evidence in a *partial image* and optionally a *text prompt*, our task is to fill in a region indicated by a *binary mask* such that the resulting complete image is plausible, of high-quality, and follows the given text prompt. An extension of the simple setting is to allow *continuous masks*, which provide additional guidance about how strong certain

image regions are allowed to be altered in the inpainting process. Continuous masks enable more sophisticated edits like image mixing and gradual changes over regions (Levin & Fried, 2023). In this work, we introduce a novel framework for conditional binary and continuous mask inpainting, achieving state-of-the-art quality.

Image inpainting has recently benefited from advances in large-scale diffusion and flow-based text-to-image (T2I) models (Chen et al., 2024; Esser et al., 2024; Black Forest Labs, 2024). However, models for high-resolution and high-quality image synthesis generate images in latent space (Rombach et al., 2022) and fine-tuning such models effectively for latent image inpainting remains a key challenge. Current approaches typically follow one of two paths: 1) specialized sampling strategies such as RePaint (Lugmayr et al., 2022), which completely neglect the masked image as strong conditioning and therefore require many steps and resampling, or 2) fine-tuning with mask concatenation (Rombach et al., 2022), which requires architecture changes and often yields suboptimal results due to a weak condition injection. These limitations become especially apparent in more challenging settings such as fast inpainting with a low number of sampling steps.

We propose Saint, inpainting with the recently introduced Spatial Reasoning Model (SRM) framework (Wewer et al., 2025), which allows considering multiple variables with varying noise levels across masked and unmasked regions during training. Using this formulation, we obtain a principled way to finetune pre-trained diffusion and rectified flow transformers (Peebles & Xie, 2023) for binary and continuous mask inpainting in latent space. Unlike previous works with two noise levels, like TD-Paint (Mayet et al., 2025), which uses a UNet trained from scratch in pixel space with a fixed noise scheme, Saint brings this concept to high-capacity transformers for latent generation and generalizes naturally to continuous mask settings. We further leverage the principle of masking via noising by introducing *Spatial Classifier-Free Guidance* and show that it effectively enhances quality and consistency with the input image. By denoising only masked regions with a transformer-based architecture, Saint enables *Clean Token Caching* to accelerate fine-grained inpainting.

We present extensive experiments and ablations, showing that our approach outperforms existing works on the standard ImageNet1k benchmark, but more importantly in text-conditional inpainting on JourneyDB (Sun et al., 2023). In summary, our contributions are as follows:

- We demonstrate how to fine-tune pre-trained latent diffusion and rectified flow transformers as SRMs, leveraging masking via noising with inpainting-specific spatial $t$-sampling.

- We propose a novel Spatial Classifier-Free Guidance formulation tailored for inpainting with SRMs, which boosts quality by enforcing alignment with the masked input.

- We introduce Clean Token Caching for transformer-based SRMs to speed up small-scale edits by denoising masked tokens only.

- In a fair comparison using the same pre-trained models, we show that Saint outperforms previous training-free and fine-tuning approaches w.r.t. visual fidelity and input alignment in class-, text-, and unconditional inpainting with binary as well as continuous masks.

## 2 RELATED WORK

We revisit existing diffusion-based methods for inpainting (Sec. 2.1), summarize general Classifier-Free Guidance (Sec. 2.2), and briefly describe Spatial Reasoning Models (Sec. 2.3).

### 2.1 DIFFUSION-BASED IMAGE INPAINTING

Image inpainting is the task of region-constrained contextual image generation given an input image and mask pair. It is one of the cornerstones of image editing with important applications like object removal or background replacement (Huang et al., 2025). While inpainting is a well-established task with a long history of approaches (Criminisi et al., 2004; Barnes et al., 2011; Yang et al., 2017; Yu et al., 2018; 2019; Suvorov et al., 2022), denoising generative models (DMs) (Ho et al., 2020; Lipman et al., 2023) are the state-of-the-art nowadays. With their exceptional ability to fit complex distributions, they enabled today's text-to-image generators (Rombach et al., 2022; Nichol et al., 2022; Esser et al., 2024; Podell et al., 2024; Black Forest Labs, 2024) that are often used as a basis for inpainting methods. We review diffusion-based inpainting methods, categorizing them into training-free and mask-conditional methods.

**Training-Free Approaches.** Adapting pre-trained large generative models for inpainting without retraining has received a lot of attention in the last years (Chung et al., 2022; Couairon et al., 2023; Kawar et al., 2022; Wang et al., 2025). Notably, Blended Diffusion (Avrahami et al., 2022; 2023) showed that a DM trained *without* mask conditioning can still inpaint by repeatedly replacing the

*unmasked* region of the current sample with the *noised* conditioning image after each step. A parallel work – RePaint (Lugmayr et al., 2022) – proposes a similar method, which introduces *resampling*, i.e., repeatedly noising and denoising at each step to improve global coherence at the cost of significant computational overhead. Differential Diffusion (Levin & Fried, 2023) extends this idea to continuous masks. They are utilized as edit strength, realized by mixing the denoised sample with a re-noised copy of the input via a time-dependent, per-pixel gate.

Note that for all of those methods, at high noise levels, the model has no information about the details of the given image – resulting in a very poor coherence to computation tradeoff. Our method fine-tunes the model to explicitly reason on the given regions according to the continuous mask, making it easy to preserve global consistency.

**Mask-Conditional Methods.** With the rise of large-scale image diffusion models, image-to-image conditioning via channel-wise concatenation was introduced and implemented in Palette (Saharia et al., 2022) and later for higher resolutions in Latent Diffusion Models (Rombach et al., 2022). While this method requires fine-tuning, it is widely used due to its effectiveness (Nichol et al., 2022; Hugging Face Diffusers team; Wang et al., 2023; Yang et al., 2023). To avoid retraining, BrushNet (Ju et al., 2025) introduces a ControlNet-like method that trains a second branch which inputs the masked features and injects its features into the frozen pre-trained model, making the training non-destructive. TD-Paint (Mayet et al., 2025) improves results by explicitly training a U-Net for binary masks encoded as per-pixel noise level $t = 0$. Due to architecture modifications, they train a model from scratch. A parallel work RAD (Kim et al., 2025), also trains with two noise levels within a sample, and suggests LoRA (Hu et al., 2022) for fine-tuning pixl-space DMs. Additionally, they explore the use of Perlin noise for better alignment with arbitrary masks during inference. Both approaches are pixel-space diffusion models, and are based on a U-Net architecture, which limits their ability to make use of the state-of-the-art, high-resolution, text-to-image diffusion and rectified flow transformers, such as Flux.1 (Black Forest Labs, 2024), SD3 (Esser et al., 2024) or PixArt-$\alpha$ (Chen et al., 2024). Saint shifts this idea to a latent-space DiT setup, for easy fine-tuning of such models, and also can operate on continuous masks, making the method more versatile.

## 2.2 Classifier-Free Guidance

Classifier-Free Guidance (CFG) (Ho & Salimans, 2021) is a method used to trade off diversity for fidelity. During training of a conditional denoising model, the condition is dropped with a certain probability to enable unconditional inference. At sampling time, both conditional and unconditional flow (or noise) estimates $u_{\text{cond}}$ and $u_{\text{unc}}$ are obtained and then combined as

$$u_{\text{cfg}} = w \cdot u_{\text{cond}} - (w - 1) \cdot u_{\text{unc}}, \tag{1}$$

where $w$ is the CFG scale. For $w > 1$, $u_{\text{cfg}}$ is pushed towards conditional inference, enforcing stronger alignment with the condition and often better quality at the cost of lower diversity. Follow-ups investigated alternatives for $u_{\text{unc}}$ such as predictions of a weaker model (Karras et al., 2024) or negative prompts (Ban et al., 2024), and applied CFG for timestep intervals (Kynkäänniemi et al., 2024). We propose a new CFG formulation which improves the consistency of the inpainted region with the observed context.

## 2.3 Spatial Reasoning Models

Spatial Reasoning Models (SRMs) (Wewer et al., 2025) describe a recently proposed framework for learning joint distributions of general *variables* in arbitrary spatial domains with denoising generative models. Unlike standard diffusion models, SRMs consider image patches as variables with individual noise levels both during training and inference.

SRMs Wewer et al. (2025) were introduced on small-scale toy problems only, such as synthetic images of Sudoku in pixel space. Saint extends SRMs to latent image inpainting, advances task-specific noise level sampling, introduces a new Spatial CFG formulation for improved image consistency, and significantly increases the inference speed using our Clean Token Caching.

## 3 Saint

In this section, we describe our method in detail, starting with the steps required to turn large-scale, pre-trained diffusion / flow transformers into SRMs tailored for inpainting in Sec. 3.1. Our novel conditioning on masked images natively enables a new Classifier-Free Guidance mechanism that we explain in Sec. 3.2. Lastly, by denoising only what actually needs to be inpainted, Saint opens up the opportunity for caching of already clean tokens, which we introduce in Sec. 3.3.

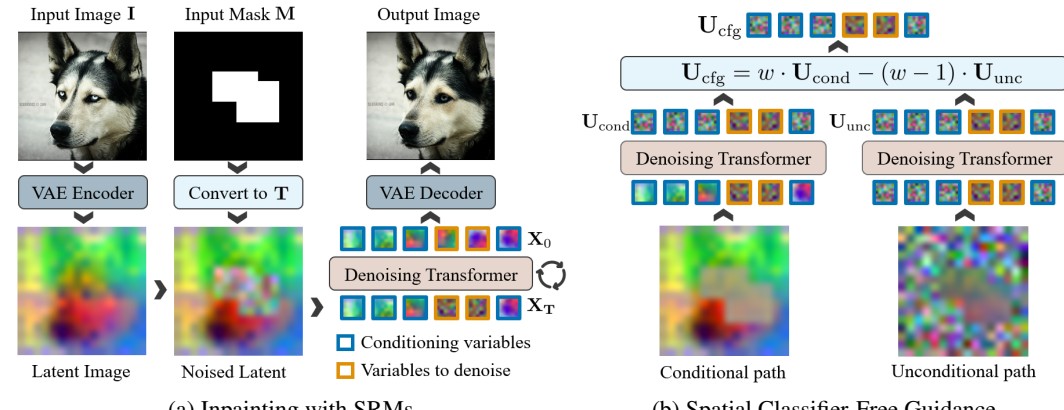

(a) Inpainting with SRMs                (b) Spatial Classifier-Free Guidance

Figure 2: **Method Overview. (a)** Saint fine-tunes latent Denoising Transformers as SRMs (Wewer et al., 2025) tailored for inpainting by converting (continuous) masks $\mathbf{M}$ into variable-wise noise levels $\mathbf{T}$ during both training and inference. **(b)** The novel inpainting procedure allows to formulate a Spatial Classifier-Free Guidance formulation, controlling the conditioning from clean variables.

### 3.1 SRMs for Inpainting

While standard flow/diffusion transformers for class- or text-conditional image generation like LightningDiT (Yao et al., 2025) or PixArt-$\alpha$ (Chen et al., 2024) are trained for sampling an entire image as a *single* random variable, SRMs (Wewer et al., 2025) aim for modeling the joint distribution of *multiple* partially observed variables. With Saint, we propose to fine-tune pre-trained DiTs as SRMs for inpainting, leveraging two natural connections: (1) DiTs tokenize the image into patches, which represent an intuitive choice for variables in the SRM framework. (2) Generative image inpainting is the task of stochastic completion of a partially observed image, which requires spatial reasoning over observed and unobserved patches. Therefore, SRMs are a perfect fit for this problem.

#### 3.1.1 Masking via Noising

To convert a pre-trained latent image DiT into a SRM, we follow the principle of masking via noising, as shown in Fig. 2a. Notably, this allows us to consider binary as well as continuous masks.

**Tokenization.** Given an image $\mathbf{I} \in \mathbb{R}^{3 \times H \times W}$ and a mask $\mathbf{M_I} \in [0, 1]^{H \times W}$, where 1 denotes a full masking, we first encode $\mathbf{I}$ to a compressed latent image $\mathbf{X} \in \mathbb{R}^{d \times h \times w}$ using the VAE encoder for latent diffusion. To match the spatial resolution, we bilinearly downsample $\mathbf{M_I}$ to obtain $\mathbf{M_X}$.

**Continuous Masking.** Next, instead of encoding the mask in the input to the denoising network via concatenation, we employ it directly as the noise level (also known as timestep) $\mathbf{T} = \mathbf{M_X}$. Leveraging the noise schedule, i.e., data and noise weighting functions $a, b$ of the diffusion/flow formulation used by the pre-trained model, the masked latent image patches $\mathbf{X^T}$ are then sampled as

$$\mathbf{X^T} = a(\mathbf{T}) \odot \mathbf{X} + b(\mathbf{T}) \odot \epsilon, \tag{2}$$

with Gaussian noise $\epsilon \sim \mathcal{N}(0, 1)$. For the popular rectified flow schedule, e.g., used by LightningDiT, this boils down to linear interpolation between the image and noise with the mask as weighting. In the case of discrete time diffusion models like PixArt-$\alpha$, we scale and discretize $\mathbf{T}$ accordingly.

**Mask Conditioning.** To condition the DiT on the mask, we utilize the trained $t$-embedding network and adaptive layer norm Peebles & Xie (2023). Both components can be adapted for token-wise noise levels $\mathbf{T}$ instead of a single shared timestep per image without introducing any additional weights.

#### 3.1.2 Training

We design a training mask sampling procedure tailored for the task of inpainting. Adapting the mask sampling from TD-Paint (Mayet et al., 2025), we first sample a binary mask, which involves sampling a patch size, a fraction of latent pixels to be masked, and finally the position of patches. This binary mask splits the latent image into conditioning variables and ones to denoise. Next, we introduce two different $t$-sampling strategies for these two sets.

For conditioning variables, we set $t = 0$ (i.e., keep them clean) with $90\%$ probability, and $t = 1$ for the remaining $10\%$ to train the denoiser for the unconditional case of Spatial CFG, which we explain in detail in Sec. 3.2. For variables to be denoised, we adapt the Uniform $\bar{t}$ strategy introduced by Wewer

et al. (2025). The strategy first samples a mean $\bar{t}$ from a uniform distribution followed by sampling a vector $\mathbf{t}$ of individual noise levels with the particular mean. Depending on the parameterization of the pre-trained model and the used SNR sampler, we replace the uniform distribution for $\bar{t}$ with, e.g., a logit normal distribution as introduced by Esser et al. (2024) for rectified flow transformers.

### 3.1.3 INFERENCE

Our inference algorithm is given in Alg. 1. After encoding the image to a latent image and the mask as noise levels (line 1) as described in Sec. 3.1.1, we start the sampling process from correspondingly noised latents in line 3. Crucially, this means that partially noised tokens in the case of a continuous mask are ahead in $t$ compared to others. During sampling, we perform equal-sized steps for every token until reaching $t = 0$. Note that while we have chosen the flow parameterization and an Euler sampler for the pseudocode, our approach is independent of this choice and can be used as well with discrete-time diffusion models. Lines 5-10 formalize our novel Spatial Classifier-Free Guidance formulation that we explain in the next Sec. 3.2.

---

**Algorithm 1** Saint Inference

**Require:** Image $\mathbf{I}$, Mask $\mathbf{M}$, Steps $n$, CFG scale $w$
1: $\mathbf{X}, \mathbf{T} \leftarrow \text{encode}(\mathbf{I}), \text{bilinear}(\mathbf{M})$
2: $\epsilon \sim \mathcal{N}(\mathbf{0}, \mathbf{1})$
3: $\mathbf{X} \leftarrow a(\mathbf{T}) \odot \mathbf{X} + b(\mathbf{T}) \odot \epsilon$
4: **for** $i = 1$ to $n$ **do**
5: $\quad \mathbf{C} \leftarrow \mathbf{T} == 0$ $\qquad \triangleright$ Clean Token Mask
6: $\quad \mathbf{T}' \leftarrow \max(\mathbf{T}, \mathbf{C})$ $\qquad \triangleright$ Unconditional $\mathbf{T}$
7: $\quad \varepsilon \sim \mathcal{N}(\mathbf{0}, \mathbf{1})$
8: $\quad \mathbf{X}' \leftarrow (1 - \mathbf{C}) \odot \mathbf{X} + \mathbf{C} \odot \varepsilon$
9: $\quad \mathbf{U}_{\text{cond}}, \mathbf{U}_{\text{unc}} \leftarrow \texttt{DiT}(\mathbf{X}, \mathbf{T}), \texttt{DiT}(\mathbf{X}', \mathbf{T}')$
10: $\quad \mathbf{U}_{\text{cfg}} \leftarrow w \cdot \mathbf{U}_{\text{cond}} - (w - 1) \cdot \mathbf{U}_{\text{unc}}$
11: $\quad \mathbf{X} \leftarrow \text{Euler}(\mathbf{X}, \mathbf{U}_{\text{cfg}}, n)$
12: $\quad \mathbf{T} \leftarrow \max(0, \mathbf{T} - 1/n)$
13: **return** $\text{decode}(\mathbf{X})$

---

### 3.2 SPATIAL CLASSIFIER-FREE GUIDANCE

By performing masking via noising with spatially varying $t$, Saint natively enables a novel classifier-free guidance formulation w.r.t. the partially given image for inpainting. Our Spatial CFG implementation is visualized in Fig. 2b and formalized in lines 5-10 of Alg. 1. Besides the conditional flow (or noise) $\mathbf{U}_{\text{cond}}$ prediction path that effectively leverages the partially given latent image as strong condition, we introduce an unconditional path $\mathbf{U}_{\text{unc}}$ for which we replace every clean input token ($t = 0$) with Gaussian noise and set the respective noise level to $t = 1$. Following the general CFG Eq. 1, we combine both paths using a specific spatial guidance scale in line 10. Notably, the set of clean tokens can grow throughout inference due to different initial $t$ determined by the possibly continuous input mask. In this case, Spatial CFG can be seen as a form of self-guidance, enforcing consistency with the history of already generated tokens.

### 3.3 CLEAN TOKEN CACHING

To speed up the iterative generation with denoising models, existing works (Wimbauer et al., 2024; Ma et al., 2024) explore caching strategies to avoid recomputing intermediate features that barely change between consecutive sampling steps. Saint offers the opportunity for a caching strategy tailored for inpainting. In particular, unlike previous methods like RePaint Lugmayr et al. (2022) or LDM Rombach et al. (2022), Saint only denoises tokens that need to be inpainted. For all conditioning variables, inference starts already at $t = 0$. Assuming that these tokens do not change significantly between consecutive sampling steps, we propose Clean Token Caching in each block of the transformer for a certain update interval, as visualized in Fig. 3. During a cache update step, we save keys and values for conditioning variables. Then, for every other (cache readout) step, the DiT only processes tokens that need to be denoised and extends the keys and values for self-attention with the current cache. This can be understood as merged self-attention between noisy tokens and cross-attention to clean tokens. As a result, the number of cached tokens and with that the potential speed-up scales inversely proportional with the inpainting mask size, especially improving the efficiency of small edits.

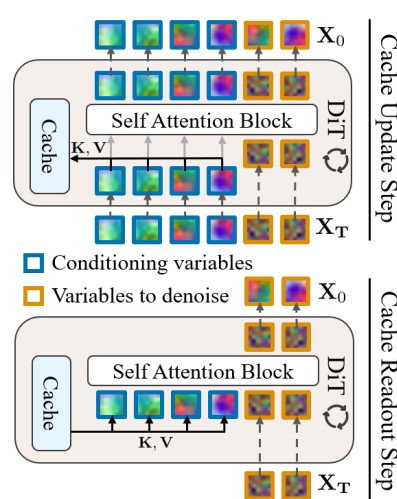

Figure 3: **Clean Token Caching.** Caching tokens for conditioning variables can speed up inpainting.

| Mask | Method | ImageNet1k 10 Steps FID↓ | LPIPS↓ | SSIM↑ | 50 Steps FID↓ | LPIPS↓ | SSIM↑ | JourneyDB 10 Steps FID↓ | LPIPS↓ | SSIM↑ | 50 Steps FID↓ | LPIPS↓ | SSIM↑ |
|---|---|---|---|---|---|---|---|---|---|---|---|---|---|
| Binary — Expand | RePaint [1] | 37.397 | 0.735 | 0.223 | 16.615 | 0.700 | 0.204 | 34.207 | 0.611 | 0.319 | 22.042 | 0.590 | 0.276 |
| | LDM [3] | 19.167 | 0.586 | 0.288 | 10.769 | 0.552 | 0.271 | 32.313 | 0.617 | 0.330 | 17.803 | 0.594 | 0.285 |
| | TD-Paint [4] | 18.780 | 0.575 | 0.294 | 10.620 | 0.540 | 0.277 | 22.955 | 0.556 | 0.357 | 13.266 | 0.539 | 0.316 |
| | **Ours** | **15.783** | **0.526** | **0.304** | **10.120** | **0.503** | **0.296** | **21.188** | **0.538** | **0.369** | **12.867** | **0.524** | **0.330** |
| Binary — Half | RePaint [1] | 13.917 | 0.382 | 0.469 | 10.008 | 0.365 | 0.465 | 19.343 | 0.346 | 0.537 | 15.218 | 0.335 | 0.519 |
| | LDM [3] | 8.290 | 0.265 | 0.535 | 6.288 | 0.250 | 0.527 | 19.226 | 0.351 | 0.541 | 15.060 | 0.342 | 0.520 |
| | TD-Paint [4] | 8.053 | 0.259 | 0.539 | 6.052 | 0.243 | 0.532 | 11.408 | 0.269 | 0.591 | 9.038 | 0.262 | 0.574 |
| | **Ours** | **7.248** | **0.237** | **0.546** | **5.869** | **0.227** | **0.543** | **9.827** | **0.246** | **0.593** | **8.690** | **0.247** | **0.586** |
| Binary — Thick | RePaint [1] | 8.778 | 0.209 | 0.603 | 7.707 | 0.206 | 0.602 | 15.287 | 0.271 | 0.606 | 10.911 | 0.242 | 0.613 |
| | LDM [3] | 4.834 | 0.126 | 0.663 | 4.186 | 0.123 | 0.657 | 18.250 | 0.235 | 0.641 | 16.872 | 0.234 | 0.630 |
| | TD-Paint [4] | 4.704 | 0.122 | 0.666 | 4.052 | 0.119 | 0.660 | 8.360 | 0.156 | 0.696 | 7.511 | 0.155 | 0.689 |
| | **Ours** | **4.437** | **0.114** | **0.669** | **3.974** | **0.113** | **0.666** | **7.734** | **0.144** | **0.702** | **7.237** | **0.145** | **0.698** |
| Continuous — Blob | Diff. Diff. [2] | 29.609 | 0.683 | 0.246 | 13.158 | 0.622 | 0.236 | 34.335 | 0.624 | 0.311 | 21.235 | 0.595 | 0.269 |
| | LDM [3] | 27.427 | 0.658 | 0.256 | 12.382 | 0.598 | 0.246 | 30.695 | 0.628 | 0.322 | 15.943 | 0.598 | 0.280 |
| | TD-Paint [4] | 18.551 | 0.571 | 0.293 | 11.317 | 0.533 | 0.279 | 59.273 | 0.655 | **0.333** | 26.885 | 0.599 | **0.305** |
| | **Ours** | **14.097** | **0.525** | **0.298** | **10.221** | **0.496** | **0.293** | **21.118** | **0.591** | 0.306 | **13.845** | **0.568** | 0.295 |
| Continuous — Gradient | Diff. Diff. [2] | 17.627 | 0.352 | 0.428 | 7.658 | 0.273 | 0.452 | 30.408 | 0.481 | 0.431 | 17.481 | 0.433 | 0.419 |
| | LDM [3] | 17.477 | 0.339 | 0.437 | 7.484 | 0.265 | 0.457 | 27.743 | 0.486 | 0.431 | 15.025 | 0.435 | 0.421 |
| | TD-Paint [4] | 8.687 | 0.259 | 0.477 | 7.131 | 0.246 | **0.473** | 36.335 | 0.462 | **0.472** | 19.714 | 0.412 | **0.466** |
| | **Ours** | **8.323** | **0.253** | **0.480** | **6.651** | **0.240** | 0.472 | **18.869** | **0.402** | 0.456 | **12.327** | **0.378** | 0.460 |
| Continuous — Soft | Diff. Diff. [2] | 10.787 | 0.241 | 0.547 | 9.152 | 0.182 | 0.589 | 15.287 | 0.271 | 0.606 | 10.911 | 0.242 | 0.613 |
| | LDM [3] | 10.291 | 0.223 | 0.561 | 5.534 | 0.171 | 0.595 | 16.460 | 0.289 | 0.592 | 11.459 | 0.250 | 0.606 |
| | TD-Paint [4] | 5.660 | 0.152 | 0.622 | 4.889 | 0.146 | 0.619 | 18.440 | 0.258 | **0.637** | 12.152 | 0.222 | 0.643 |
| | **Ours** | **5.283** | **0.147** | **0.624** | **4.632** | **0.142** | 0.620 | **10.885** | **0.211** | 0.637 | **8.496** | **0.196** | **0.644** |

Table 1: **Quantitative Class-/Text-Conditional Inpainting Comparison.** Saint consistently achieves higher fidelity and alignment with the input image. The largest advantage over baselines is obtained for low number of sampling steps (10) and / or continuous masks.

# 4 EXPERIMENTS

In this section, we describe our extensive experimental evaluation of Saint. After an introduction of the experimental setup in Sec. 4.1, we provide qualitative and quantitative evidence for the following claims: Saint (Sec. 4.2) outperforms previous methods for class- and text-conditional inpainting in terms of visual quality as well as consistency with the original image, (Sec. 4.3) enables editing with continuous masks, a setting in which baselines severely lack quality, and (Sec. 4.4) maintains strong performance even in the unconditional case. Finally, Sec. 4.5 demonstrates the effect of the proposed Spatial Classifier-Free Guidance and Sec. 4.6 shows that Clean Token Caching significantly speeds up sampling with minimal to no loss of quality.

## 4.1 EXPERIMENTAL SETUP

We defined an experimental setup for fair comparison with baselines by starting from the same pre-trained models and fine-tuning on the same datasets. Appendix A provides additional details.

**Pre-Trained Models.** Our experiments are performed using two pre-trained DiTs. LightningDiT (XL) (Yao et al., 2025) is a recently proposed, improved DiT architecture trained as a rectified flow model for class-conditional generation on ImageNet1k. PixArt-$\alpha$ (Chen et al., 2024) is an original DiT trained as a classical discrete-time diffusion model for open-vocabulary text-to-image generation.

**Datasets.** As we fine-tune for a new task, but not for a new domain, we choose ImageNet1k (Russakovsky et al., 2015) as the training dataset for LightningDit and JourneyDB (Sun et al., 2023) for PixArt-$\alpha$, both in $256 \times 256$ resolution. The latter is a dataset of 4 million image-prompt pairs generated with Midjourney and has been also used in the original training of PixArt-$\alpha$. We use the same test set of 5k images as TD-Paint (Mayet et al., 2025) for ImageNet1k and create an equally sized subset of JourneyDB's test split ensuring no prompt duplicates.

**Masks.** We adopt the binary masks *Expand*, *Half*, and *Thick* (aka *Wide*) from Mayet et al. (2025). Furthermore, we introduce continuous mask equivalents *Blob*, *Gradient*, and *Soft*. Please see Fig. 1 for visualizations and the Appendix A for details about their creation.

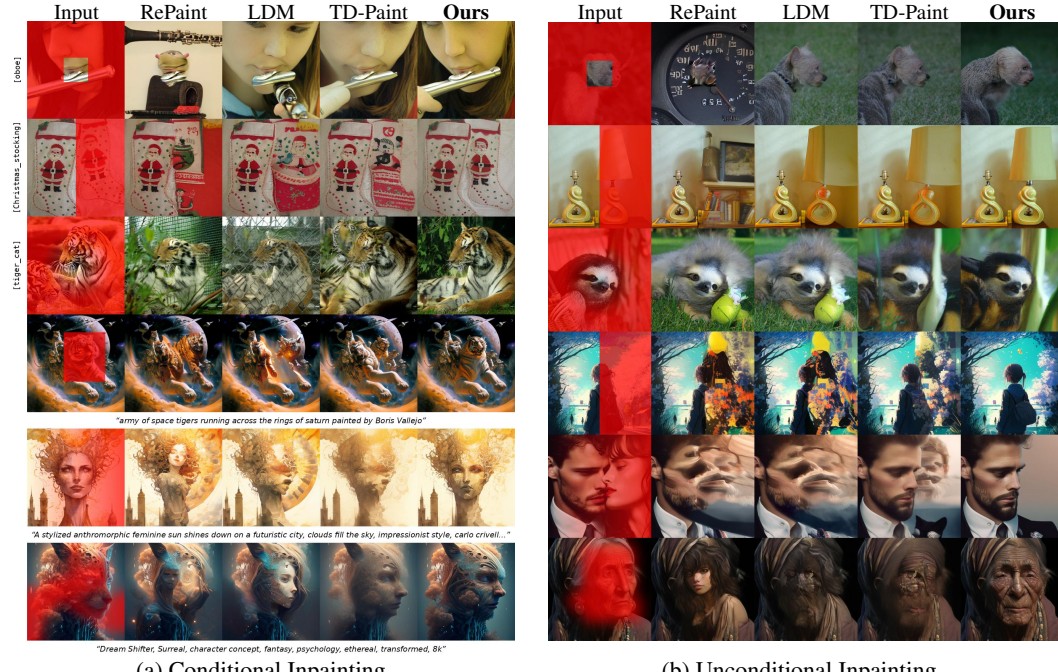

(a) Conditional Inpainting                    (b) Unconditional Inpainting

Figure 4: **Qualitative Comparison.** **(a)** Class-/Text-Conditional inpainting with Saint achieves higher quality and consistency with the masked input image. From first to last row: It successfully generates a correct head pose fitting observed lips, matching Christmas stockings, more accurate tiger bodies, and artistic face completions adhering to the original style. **(b)** Our approach is more robust than baselines in the unconditional setting. For example, it correctly outpaints a monkey (instead of giving in to the dog bias in ImageNet) and inpaints more plausible faces.

**Sampling.** Image inpainting is usually used in interactive editing scenarios Huang et al. (2025). Therefore, we focus on fast sampling settings with a low number of $10$ or $50$ denoising steps. For class- and text-conditional inpainting, we use a corresponding CFG scale of $1.5$ for all methods.

**Baselines.** We use the same pre-trained models for Saint and all baselines. RePaint (Lugmayr et al., 2022) and Differential Diffusion (Levin & Fried, 2023) use the original models without fine-tuning, as those have been already trained to convergence. LDM (Rombach et al., 2022) with mask concatenation and a latent-diffusion adaptation of TD-Paint (Mayet et al., 2025) are fine-tuned for 100k iterations, matching Saint. Since (to the best of our knowledge) no method other than Differential Diffusion focuses on the task of inpainting with continuous masks, we adapt TD-Paint and LDM for this setting. For LDM, we threshold continuous masks at t=0, as it requires binary removal of masked regions. As a result, some information is lost, but the input is guaranteed to be in the training distribution. While trained for binary masks only, latent TD-Paint directly uses continuous masks via noising similar to Saint.

**Metrics.** For quantitative comparison, we employ FID (Heusel et al., 2017) to measure visual quality. Furthermore, the established reconstruction metrics LPIPS (Zhang et al., 2018) and SSIM (Wang et al., 2004) are used to evaluate consistency with the original image.

### 4.2 High-Quality Conditional Inpainting

We provide quantitative comparisons for class-/text-conditional inpainting in Tab. 1. Saint consistently outperforms all baselines in visual quality and reconstruction, for both 10 and 50 sampling steps. Notably, for the binary mask setting and 10 steps, we improve on the best baseline – our latent TD-Paint – by $10.54\%$ / $9.68\%$ in FID and $7.86\%$ / $6.49\%$ in LPIPS on ImageNet1k and JourneyDB, respectively. Fig. 4a shows qualitative comparisons. Our approach achieves significantly improved consistency with the partially given image, visible in the form of the correct head pose fitting the observed lips in the first row, or matching Christmas stockings in the second. The same holds for the lower three examples of text-conditional inpainting, for which Saint delivers high-quality samples, while baselines lack visual fidelity or coherence with the given image regions.

| Mask | | Method | ImageNet1k | | | | | | JourneyDB | | | | | |
|---|---|---|---|---|---|---|---|---|---|---|---|---|---|---|
| | | | 10 Steps | | | 50 Steps | | | 10 Steps | | | 50 Steps | | |
| | | | FID↓ | LPIPS↓ | SSIM↑ | FID↓ | LPIPS↓ | SSIM↑ | FID↓ | LPIPS↓ | SSIM↑ | FID↓ | LPIPS↓ | SSIM↑ |
| Binary | Expand | RePaint [1] | 100.822 | 0.781 | 0.212 | 56.404 | 0.741 | 0.188 | 73.066 | 0.697 | 0.266 | 44.473 | 0.670 | 0.214 |
| | | LDM [3] | 31.182 | 0.599 | 0.288 | 18.317 | 0.561 | 0.269 | 66.021 | 0.700 | 0.274 | 38.355 | 0.670 | 0.220 |
| | | TD-Paint [4] | 29.677 | 0.590 | 0.294 | 17.284 | 0.551 | 0.275 | 41.180 | 0.636 | 0.307 | 23.735 | 0.611 | 0.256 |
| | | **Ours** | **21.227** | **0.535** | **0.295** | **14.096** | **0.509** | **0.293** | **26.182** | **0.587** | **0.351** | **16.185** | **0.574** | **0.307** |
| | Half | RePaint [1] | 30.821 | 0.407 | 0.459 | 27.144 | 0.389 | 0.454 | 39.144 | 0.388 | 0.513 | 30.742 | 0.374 | 0.491 |
| | | LDM [3] | 10.119 | 0.267 | 0.537 | 7.577 | 0.250 | 0.529 | 35.013 | 0.380 | 0.537 | 30.414 | 0.370 | 0.511 |
| | | TD-Paint [4] | 9.727 | 0.261 | 0.540 | 7.207 | 0.244 | 0.533 | 13.128 | 0.284 | 0.582 | 10.294 | 0.276 | 0.565 |
| | | **Ours** | **8.205** | **0.238** | **0.542** | **6.814** | **0.226** | **0.543** | **10.165** | **0.251** | **0.588** | **9.274** | **0.251** | **0.584** |
| | Thick | RePaint [1] | 15.651 | 0.220 | 0.598 | 15.015 | 0.217 | 0.597 | 20.074 | 0.230 | 0.648 | 16.690 | 0.228 | 0.640 |
| | | LDM [3] | 5.444 | 0.126 | 0.664 | 4.668 | 0.123 | 0.658 | 27.278 | 0.246 | 0.639 | 25.147 | 0.244 | 0.626 |
| | | TD-Paint [4] | 5.279 | 0.123 | 0.667 | 4.496 | 0.120 | 0.661 | 9.107 | 0.162 | 0.693 | 8.096 | 0.161 | 0.686 |
| | | **Ours** | **4.797** | **0.114** | **0.669** | **4.344** | **0.111** | **0.669** | **7.896** | **0.148** | **0.699** | **7.497** | **0.148** | **0.696** |
| Continuous | Blob | Diff. Diff. [2] | 72.589 | 0.725 | 0.238 | 44.328 | 0.658 | 0.225 | 73.832 | 0.711 | 0.264 | 38.399 | 0.673 | 0.211 |
| | | LDM [3] | 50.033 | 0.684 | 0.259 | 26.779 | 0.618 | 0.245 | 65.442 | 0.717 | 0.264 | 34.029 | 0.678 | 0.214 |
| | | TD-Paint [4] | 30.794 | 0.592 | 0.294 | 18.732 | 0.548 | 0.278 | 99.294 | 0.752 | **0.296** | 55.114 | 0.683 | **0.258** |
| | | **Ours** | **21.109** | **0.537** | **0.296** | **14.827** | **0.506** | **0.291** | **29.234** | **0.655** | 0.256 | **20.827** | **0.640** | 0.240 |
| | Gradient | Diff. Diff. [2] | 22.883 | 0.359 | 0.429 | 9.629 | 0.277 | 0.452 | 62.001 | 0.538 | 0.413 | 32.520 | 0.479 | 0.395 |
| | | LDM [3] | 21.863 | 0.343 | 0.439 | 9.220 | 0.267 | 0.457 | 60.865 | 0.547 | 0.409 | 32.985 | 0.484 | 0.394 |
| | | TD-Paint [4] | 10.648 | 0.261 | 0.479 | 8.526 | 0.247 | **0.474** | 59.169 | 0.512 | **0.457** | 29.624 | 0.451 | **0.450** |
| | | **Ours** | **10.006** | **0.255** | **0.480** | **7.809** | **0.241** | 0.472 | **21.283** | **0.427** | 0.441 | **15.107** | **0.405** | 0.447 |
| | Soft | Diff. Diff. [2] | 15.134 | 0.248 | 0.546 | 7.961 | 0.187 | 0.588 | 23.529 | 0.294 | 0.597 | 15.980 | 0.261 | 0.603 |
| | | LDM [3] | 13.111 | 0.226 | 0.562 | 6.770 | 0.173 | 0.595 | 28.647 | 0.316 | 0.584 | 18.947 | 0.270 | 0.597 |
| | | TD-Paint [4] | 6.620 | 0.154 | 0.622 | 5.637 | 0.147 | 0.619 | 22.033 | 0.275 | 0.632 | 14.059 | 0.235 | 0.639 |
| | | **Ours** | **6.061** | **0.146** | **0.624** | **5.233** | **0.142** | **0.622** | **11.266** | **0.217** | **0.633** | **9.271** | **0.202** | **0.643** |

Table 2: **Quantitative Unconditional Inpainting Comparison.** Our method attains more robust performance in the unconditional setting than baselines. Even though Saint fine-tunes pre-trained DiTs that heavily rely on class or text conditions, our effective masking via noising together with Spatial CFG result in a smaller gap to conditional inpainting.

### 4.3 FLEXIBLE CONTINUOUS MASKING

While previous methods like Diff. Diff. (Levin & Fried, 2023) either do not leverage the input image as conditioning at all or assume only binary masks (TD-Paint (Mayet et al., 2025)), Saint enables flexible inpainting with continuous masks. As a result, our method significantly outperforms all baselines in this setting as shown in Tab. 1. We found the fine-tuning with spatially varying noise levels to be essential for the text-conditional PixArt-$\alpha$, which fails entirely when tuned for binary masks only as for our latent TD-Paint baseline. On JourneyDB, Saint improves FID compared to existing works by 30.66% with 10 steps, even outperforming all baselines with five times as many steps for soft masks. Similar observations can be made for unconditional inpainting. Qualitatively, we observe that higher mask values (higher noise) leads to significant structural changes, while lower mask values only allow changes in high frequency, textural content.

### 4.4 ROBUST UNCONDITIONAL INPAINTING

Real-world inpainting applications like object removal or occlusion recovery usually lack a class or text conditioning as input. To evaluate robustness for this use case, we leverage the same fine-tuned models, which have seen the unconditional case for 10% of all training samples. The quantitative comparison is given in Tab. 2. Saint is more robust than all baselines. By dropping the extra conditioning, the average FID over all mask settings and for 10 steps increases by only 23.42% on ImageNet1k and 13.93% on JourneyDB for our approach, compared to 32.76% / 42.21% for the best baseline TD-Paint. Even in the unfair comparison of Saint with only 10 steps versus baselines with 50 steps, Saint wins for almost all masks on JourneyDB. Looking at qualitative comparisons in Fig. 4b, our method produces high-fidelity samples coherent with the masked input, without the need for any extra conditioning. In contrast, RePaint / Diff. Diff. fails for almost all input masks due to the low number of sampling steps, which does not allow the model to correct the initially uninformed predictions. Overall, our method leverages the partially given image much more effectively and with that maintains strong performance in the unconditional setting.

### 4.5 BOOSTED QUALITY WITH SPATIAL CFG

Fig. 5 visualizes the relative performance gain in terms of FID and LPIPS for different Spatial CFG scales. Note that scale 1 is equivalent to no Spatial CFG. Across all datasets, conditioning settings,

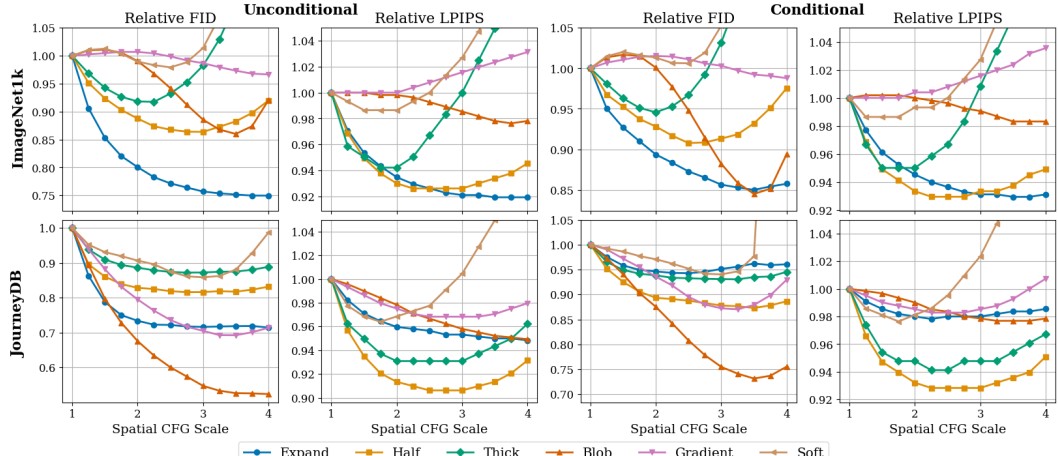

Figure 5: **Spatial CFG Scale Ablation.** We visualize the relative improvements in FID and LPIPS for 10 steps, when we increase the Spatial CFG weight. While it boosts metrics in almost all settings, it is especially beneficial for high mask ratios (*Expand*, *Half*, *Blob*) and / or unconditional inpainting. In these cases of weak conditioning, it pushes inference to better leverage the masked input image.

and inpainting masks, Spatial CFG consistently improves performance. It proves to be especially effective for large masking ratios like *Expand* and *Blob*, reducing FID by up to $15.52\%$ / $26.85\%$ for class-/text-conditional inpainting on ImageNet1k and JourneyDB, respectively. Furthermore, it can compensate, to some extent, for the absence of guidance w.r.t. additional conditioning signals, as can be seen in the form of even larger FID gains of up to $25.05\%$ / $47.60\%$ in the unconditional setting. Overall, Spatial CFG boosts the inpainting quality, especially for a small number of steps (10). We provide an additional analysis for 50 steps as well as for CFG composition w.r.t. other conditions in Appendix B, and show a qualitative ablation in Appendix C.

### 4.6 FASTER INPAINTING WITH CLEAN TOKEN CACHING

Leveraging the DiT architecture to denoise only tokens that need to be inpainted, we proposed Clean Token Caching (CTC) in Sec. 3.3. Fig. 6 shows the quality-throughput trade-off for different cache update intervals in terms of FID and number of inpainted images per second, relative to without it, i.e., a cache update interval of 1 step. We ensure comparable results using a NVIDIA L40 GPU for 50 sampling steps with a fixed batch size of 128 / 32 for ImageNet1k and JourneyDB, respectively. CTC can significantly speed up inpainting with negligible loss of quality. Depending on the mask sizes, we achieve speed-ups of up to $156.53\%$ / $109.19\%$ with at most $1.31\%$ / $0.26\%$ increase of FID for a cache update interval of 10 steps. This shows that clean ($t = 0$) tokens in each transformer block do not change significantly between consecutive sampling steps, despite full attention between all tokens. CTC exploits this and avoids unnecessary recomputation. As a result, Saint is especially efficient for small edits.

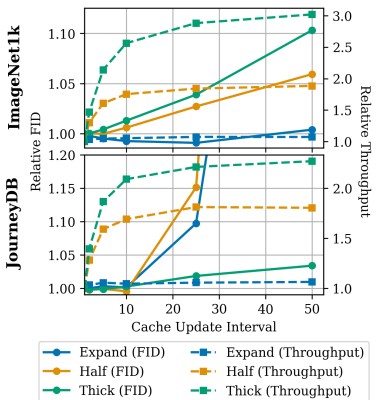

Figure 6: **CTC Speedup.** Clean Token Caching speeds up inpainting with almost no loss of quality loss.

## 5 CONCLUSION

We presented Saint, a novel approach to fine-tuning latent Diffusion Transformers for image inpainting. By performing masking via noising with the mask as noise levels, we enable flexible inpainting with continuous masks. The proposed Spatial CFG boosts quality by guiding the inference process to fully leverage the masked input image as conditioning. Clean Token Caching speeds up sampling with negligible loss of quality by denoising only what needs to be inpainted. Together, Saint achieves state-of-the-art class-, text-, and unconditional inpainting.

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

# Saint: Spatial Guidance for Inpainting

## Appendix

In this supplementary material, we provide additional experimental details including hyperparameters in Sec. A and analyze the effect of our proposed Spatial CFG for more sampling steps and in combination with CFG for other conditions in Sec. B. Furthermore, Sec. C and Sec. D provide a qualitative ablation of Spatial CFG and additional qualitative comparisons with baselines, respectively. The usage of Large Language Models (LLMs) for this paper is detailed in Sec. E.

## A  EXPERIMENTAL DETAILS

We provide experimental details in addition to the setup explained in Sec. 4.1.

**Continuous Masks.** We create continuous mask equivalents to the binary masks from Mayet et al. (2025). *Blob* for soft outpainting is created as a radial, linear gradient from no to full masking with a radius equal to a quarter of the image diagonal. *Gradient* is a continuous version of *Half* in form of a linear gradient from no masking for the first pixel column on the left side to full masking on the right. *Soft* masks are created by applying Gaussian blur with $\sigma = 16$ on the binary *Thick* (aka Wide) masks.

**Pre-Trained Models.** We use LightningDiT (Yao et al., 2025) (XL) with 675M and PixArt-$\alpha$ (Chen et al., 2024) with 0.6B parameters as base models, both trained for $256 \times 256$ image resolution.

**Training.** Fine-tuning for Saint and baselines is done for 100k iterations and effective batch sizes 512 / 256 on 4 NVIDIA H100s for up to 2 days. We use the AdamW optimizer with learning rates $1 \times 10^{-4}$ / $5 \times 10^{-6}$, $\beta_1, \beta_2$ $(0.9, 0.95)$ / $(0.9, 0.999)$, and weight decay 0 / 0.03, following the original work by Yao et al. (2025) and established fine-tuning setups for text-to-image models.

**Sampling.** We employ the established discrete Euler and DDIM (Song et al., 2021) samplers for the flow-based LightningDiT and diffusion-based PixArt-$\alpha$, respectively. Furthermore, for inpainting on ImageNet1k, we use the same timestep shift of 0.3 as Yao et al. (2025).

## B  SPATIAL CFG ANALYSIS

This section provides an ablation of the impact of our Spatial Classifier-Free Guidance formulation, described in detail in Sec. 3.2. Tab. B1 reports FID and LPIPS scores of Saint with and without Spatial CFG for class- and text-conditional inpainting, averaged for binary and continuous masks. Spatial CFG consistently boosts both visual quality as well as alignment with the masked input image. Fig. B1 shows the relative FID gains using different Spatial CFG scales for inpainting with 50 steps for all combinations of masks, datasets, and additional conditioning. There, we observe that inpainting with binary masks – *Expand*, *Half* and *Thick* – often benefit more from the Spatial CFG compared to continuous masks. Note that at the beginning of inference with continuous masks like *Blob* or *Gradient*, almost all tokens contain at least some noise. Additionally, the fully clean region is often surrounded by tokens with at least some noise. This may lead to minimal differences between $u_{cond}$ and $u_{uncond}$, which, as we hypothesize, could result in the CFG extrapolating in a seemingly random direction and harming the quality. Overall, compared to the results for 10 steps in Fig. 5, inference with 50 steps requires lower Spatial CFG scales.

We further analyze Spatial CFG in combination with class and text CFG in Fig. B2. Using 10 sampling steps for inpainting *Expand* and *Blob* masks, we investigate relative FID gains over different CFG scales for two options. On the left, we provide results for one fully conditional and one fully

| | | ImageNet1k | | | | JourneyDB | | | |
| | | 10 Steps | | 50 Steps | | 10 Steps | | 50 Steps | |
| Mask | Method | FID↓ | LPIPS↓ | FID↓ | LPIPS↓ | FID↓ | LPIPS↓ | FID↓ | LPIPS↓ |
|---|---|---|---|---|---|---|---|---|---|
| Bin. | Ours w/o Spatial CFG | 10.416 | 0.314 | 6.837 | 0.295 | 13.980 | 0.323 | 9.845 | 0.314 |
| | **Ours** | **9.156** | **0.292** | **6.654** | **0.281** | **12.916** | **0.309** | **9.598** | **0.305** |
| Con. | Ours w/o Spatial CFG | 10.097 | 0.311 | **7.168** | **0.293** | 20.663 | 0.408 | 11.691 | **0.380** |
| | **Ours** | **9.234** | **0.308** | **7.168** | **0.293** | **16.957** | **0.401** | **11.556** | 0.381 |

Table B1: **Quantitative Spatial CFG Ablation.** We provide average scores for class-/text-conditional inpainting with binary and continuous masks with and without Spatial CFG.

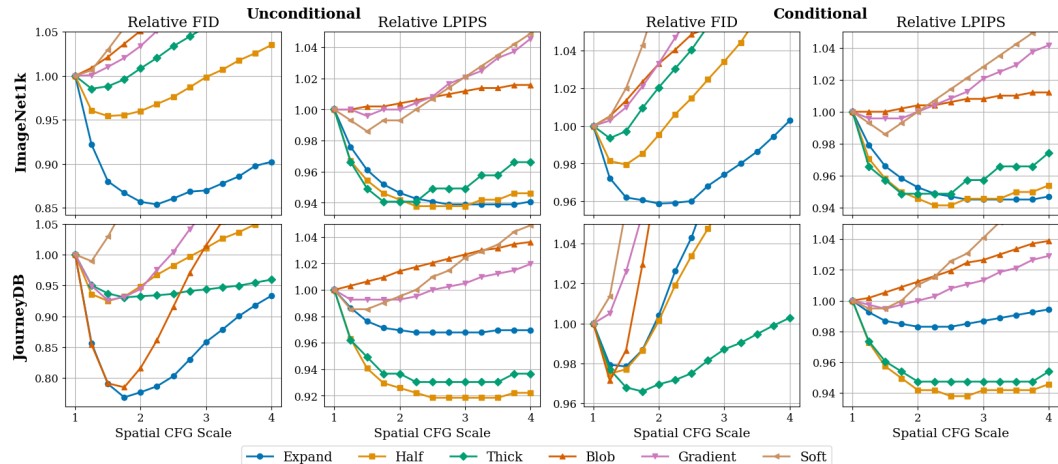

Figure B1: **Spatial CFG Scale Ablation for 50 Steps.** Compared to the same ablation for 10 steps in Fig. 5, inference with more steps benefits from lower Spatial CFG scales.

unconditional forward pass with predictions combined using a single shared CFG scale. On the right, we show compositional CFG (Liu et al., 2022) by combining a fully unconditional pass with two partially conditional forward passes for each condition (masked input and class label or text prompt). We can observe that the best performance is achieved using similar CFG scales for both conditions. As a result, shared CFG yields strong performance with only two forward passes. However, the strongest FID gains can be obtained by carefully tuning the individual scales in a compositional formulation. More importantly, looking at the first column of the heatmaps, i.e., scores with class or text CFG only, we can see that a combination with our Spatial CFG always improves sample quality. Therefore, we conclude that Spatial CFG adds another dimension of guidance that enforces consistency with the masked input image.

## C  QUALITATIVE SPATIAL CFG ABLATION

We provide a qualitative Spatial CFG ablation in Fig. C1 and Fig. C2 for ImageNet1k and JourneyDB, respectively. It boosts alignment with the masked input image, which in turn also improves visual fidelity. Fig. C1 supports this claim with examples being (from top to bottom) 1) finding a pose of a butterfly fitting the given image patch, 2) replicated patterns from the hair slide, 3) inpainting matching desserts, and 4) picking up the graffiti pattern on a freight car. Furthermore, it provides substantial gains in the unconditional setting, visible in the form of 5) sampling a lizard species closer to the one in the original image and 6) matching lamp stands. Similar observations can be made for text-conditional inpainting in Fig. C2. The examples show 1) more visually appealing soft resampling of the original, as well as 2) and 4) exploiting symmetries to achieve higher quality. Finally, 5) shows that our Spatial CFG can significantly help in scenarios with no text/class conditioning and a high mask ratio, where low-step inference without CFG fails due to a large number of possible samples.

## D  ADDITIONAL QUALITATIVE RESULTS

We provide extensive qualitative results in Fig. D1 to Fig. D8 with one figure per combination of dataset, conditional or unconditional inpainting, and binary or continuous masks.

## E  USE OF LARGE LANGUAGE MODELS

Large Language Models (LLMs) have been used during the work on this paper for:

- Search for relevant papers – by `Perplexity.ai`
- Feedback on the readability and clarity of the written text – by `ChatGPT`

All research contributions like the ideas for methods, their implementation, figures, experimental evaluation, and analyses are solely our own work.

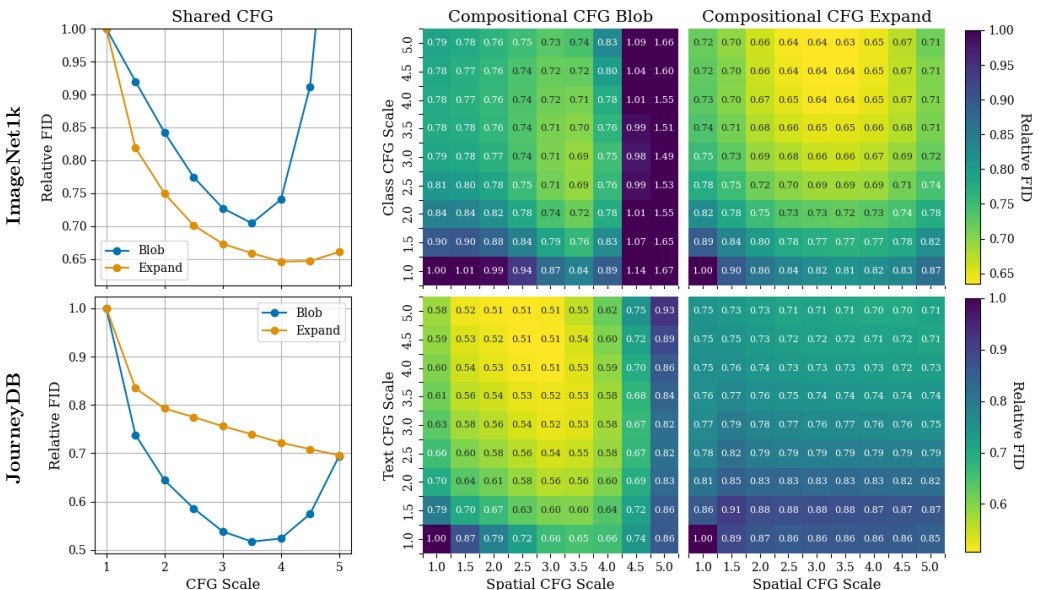

Figure B2: **CFG Composition Analysis.** We analyze the combination of our Spatial CFG with common class label or text prompt CFG in terms of relative FID gains for the two masks *Blob* and *Expand* and 10 sampling steps. Left: CFG with a shared scale. Right: grid search of individual scales. Comparing the first columns in the heatmaps with the corresponding best scores reveals that Spatial CFG cannot simply be replaced with a higher scale for class or text CFG, but supplements guidance along an additional dimension: consistency with the masked input.

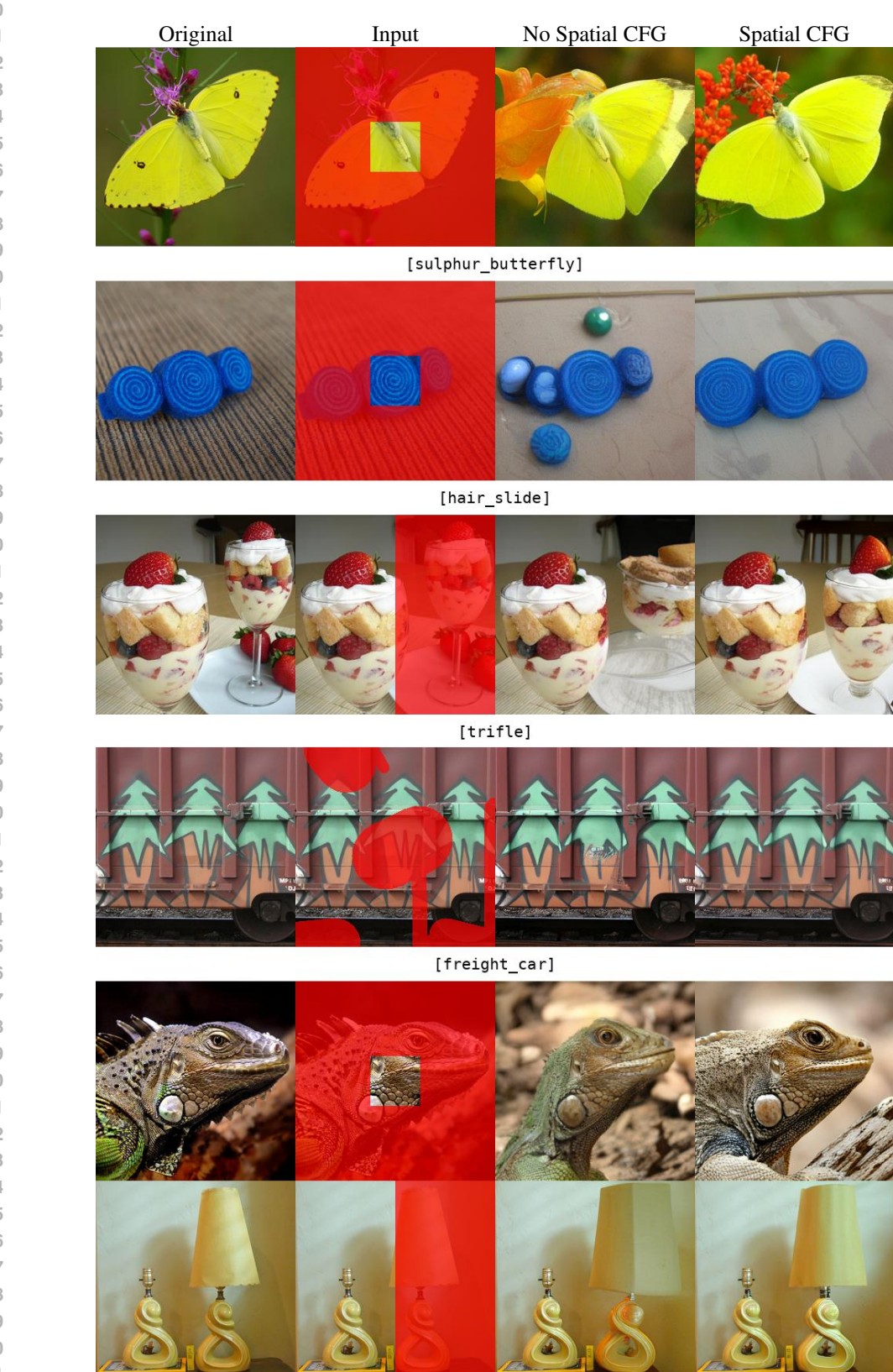

Figure C1: **Qualitative Spatial CFG Ablation on ImageNet1k.**

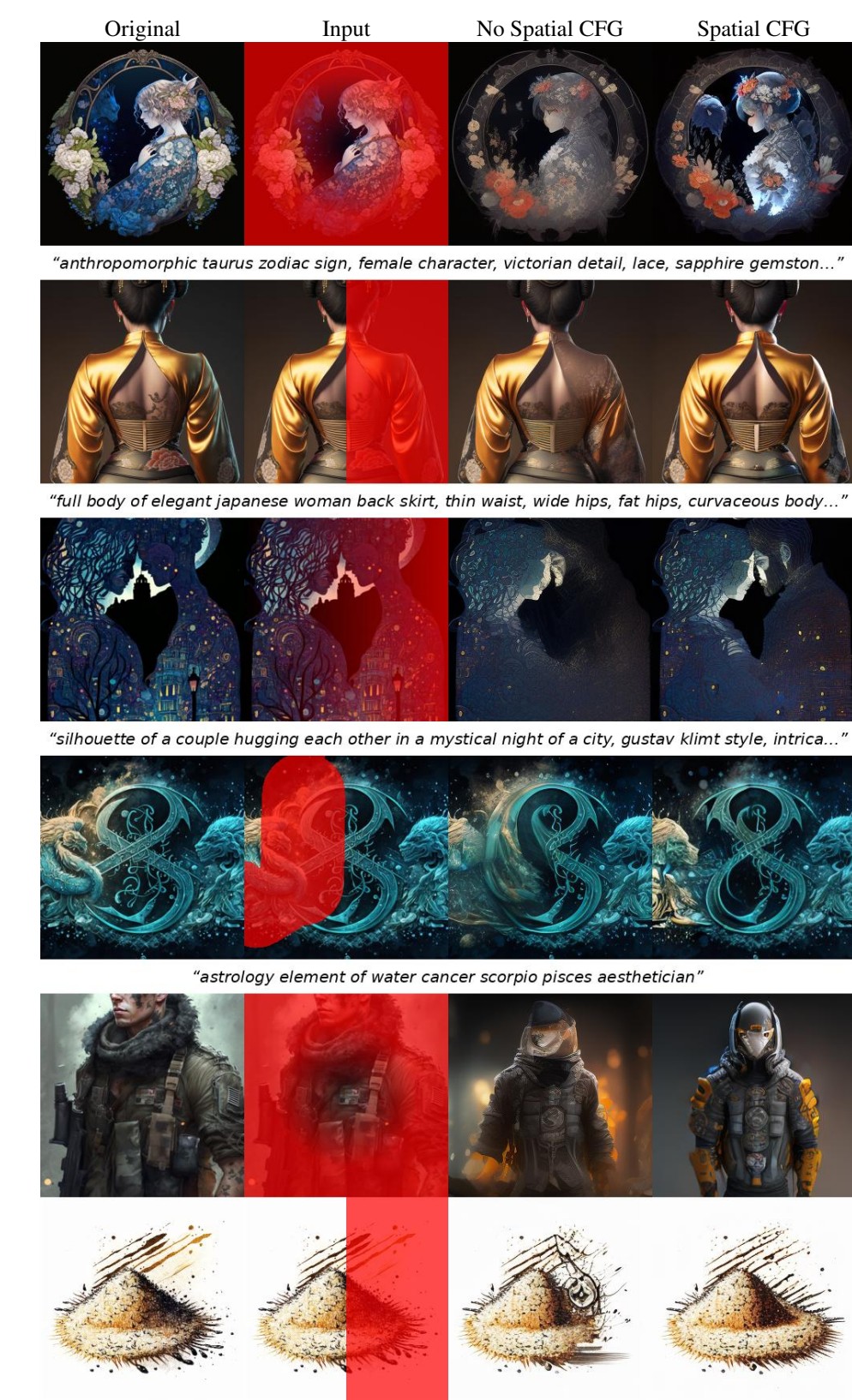

Figure C2: **Qualitative Spatial CFG Ablation on JourneyDB.**

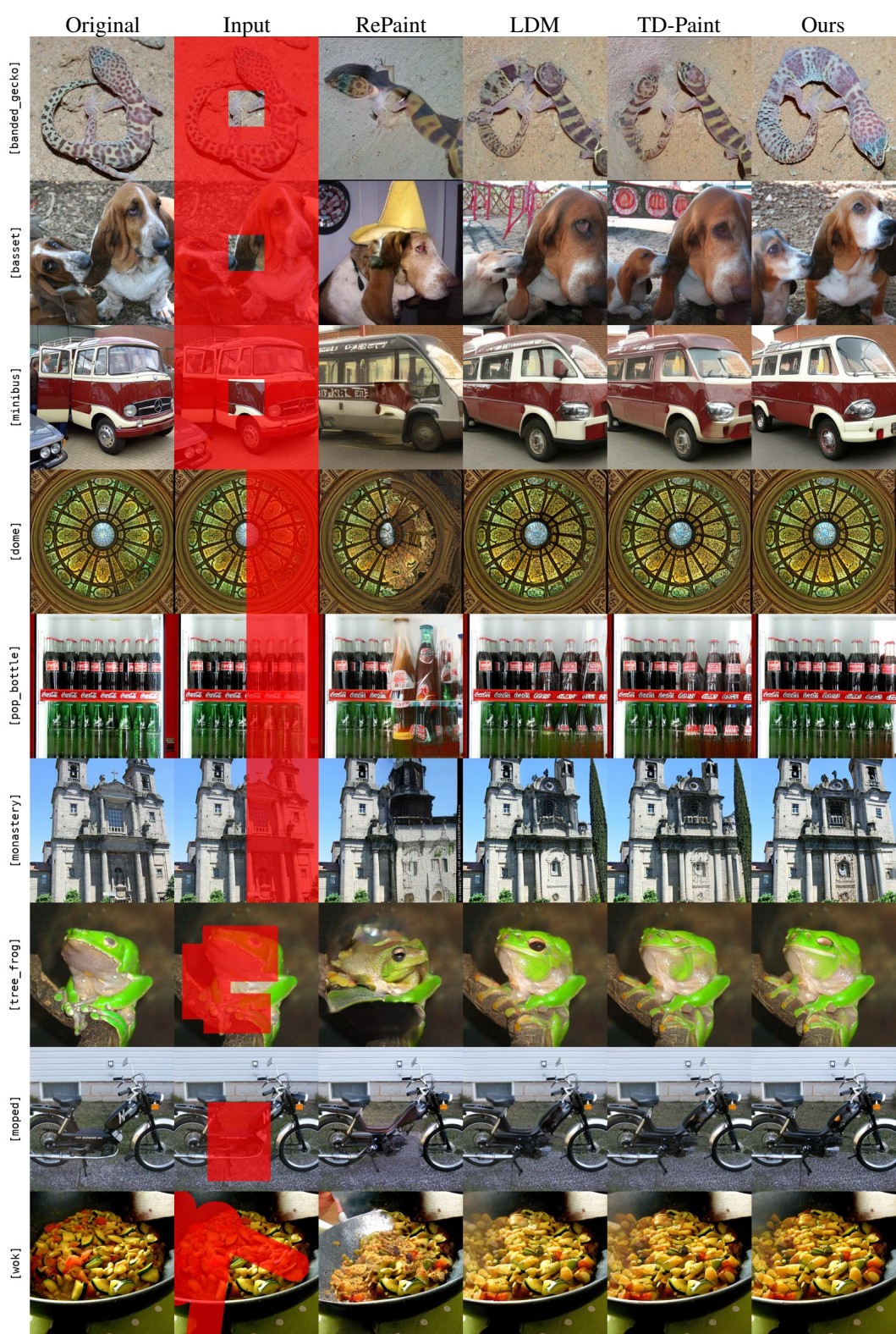

Figure D1: **Class-Conditional Inpainting on ImageNet1k with Binary Masks.**

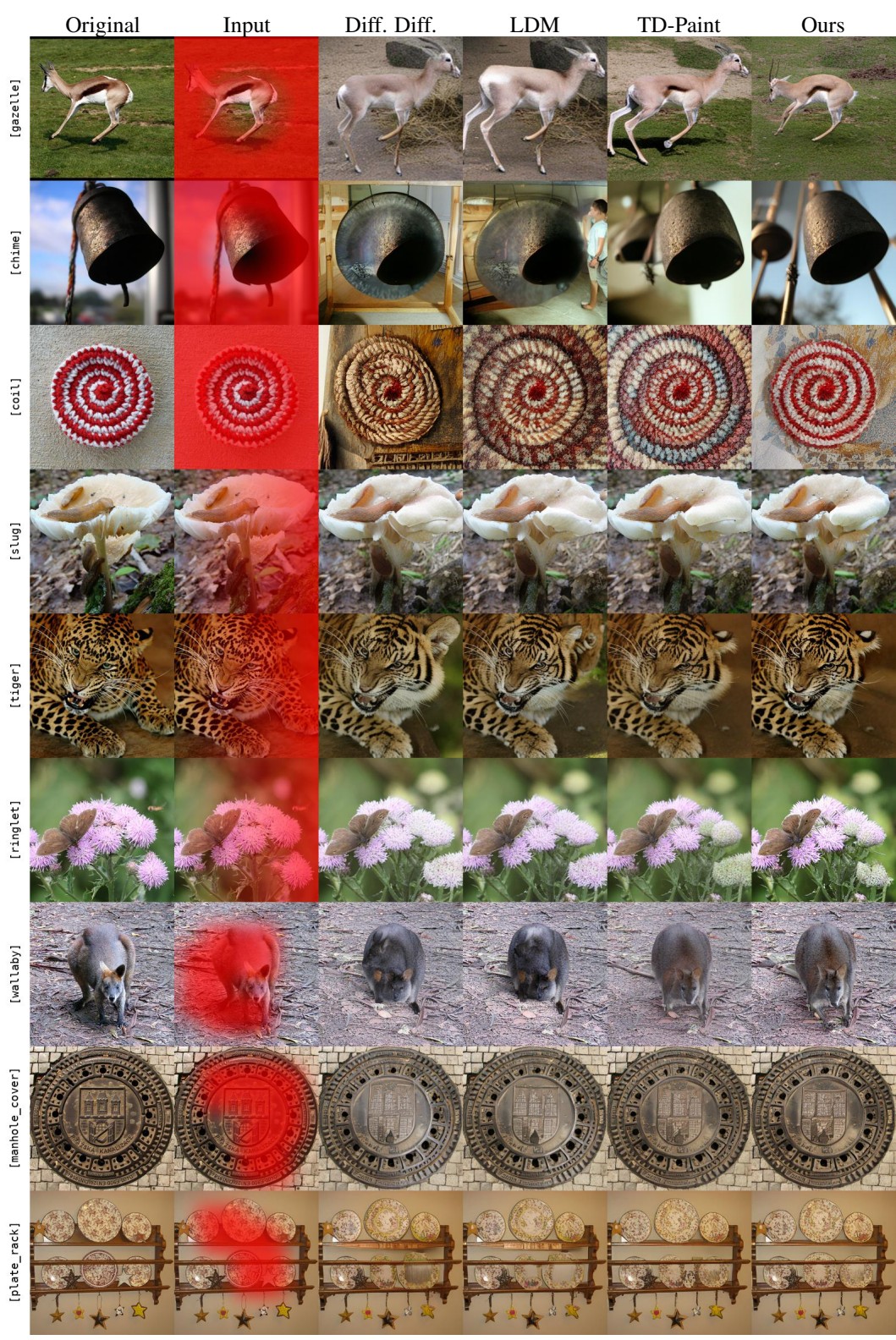

Figure D2: **Class-Conditional Inpainting on ImageNet1k with Continuous Masks.**

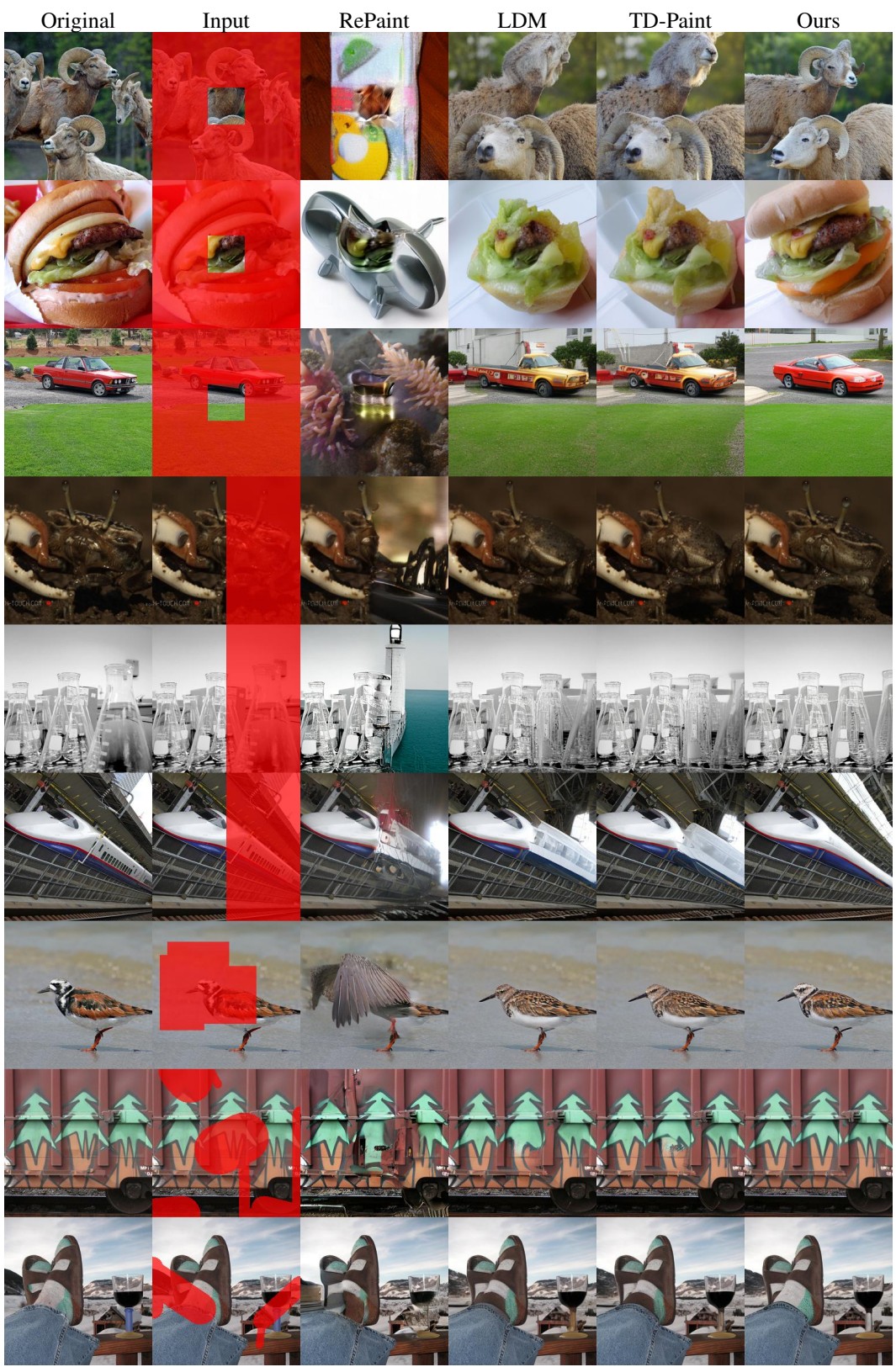

Figure D3: **Unconditional Inpainting on ImageNet1k with Binary Masks.**

| Original | Input | Diff. Diff. | LDM | TD-Paint | Ours |
|----------|-------|-------------|-----|----------|------|

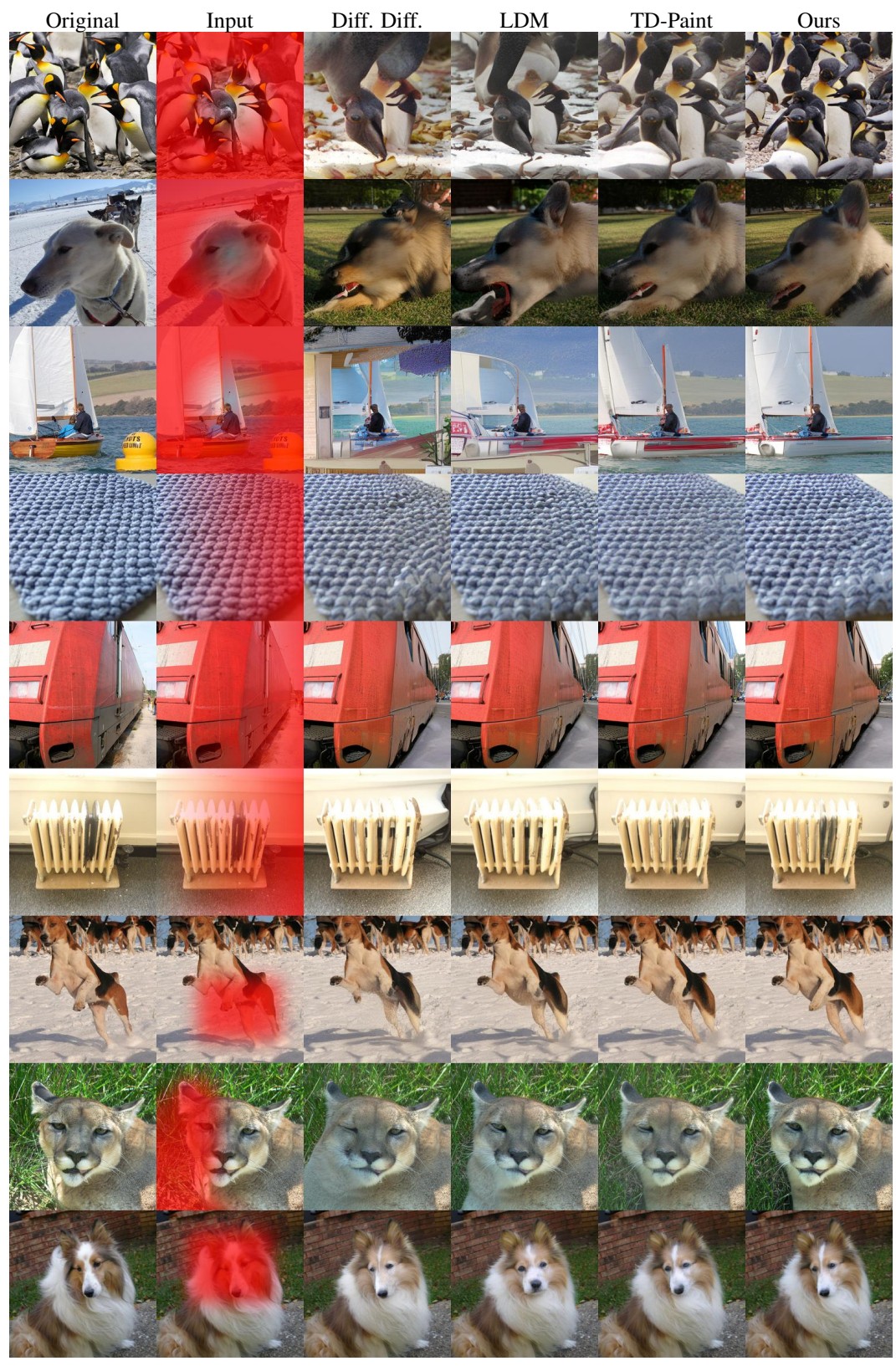

Figure D4: **Unconditional Inpainting on ImageNet1k with Continuous Masks.**

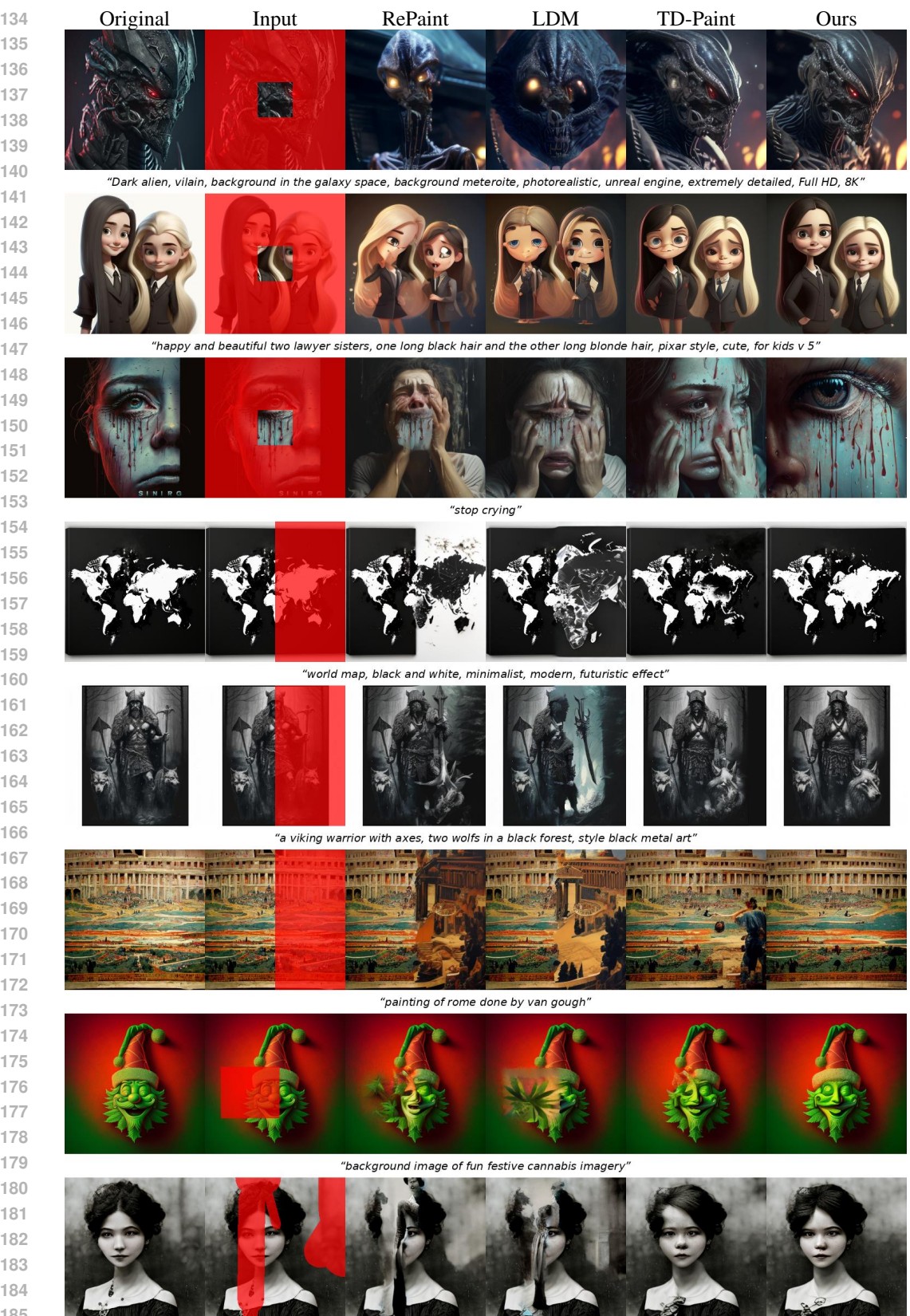

Figure D5: **Text-Conditional Inpainting on JourneyDB with Binary Masks.**

| Original | Input | Diff. Diff. | LDM | TD-Paint | Ours |
|----------|-------|-------------|-----|----------|------|

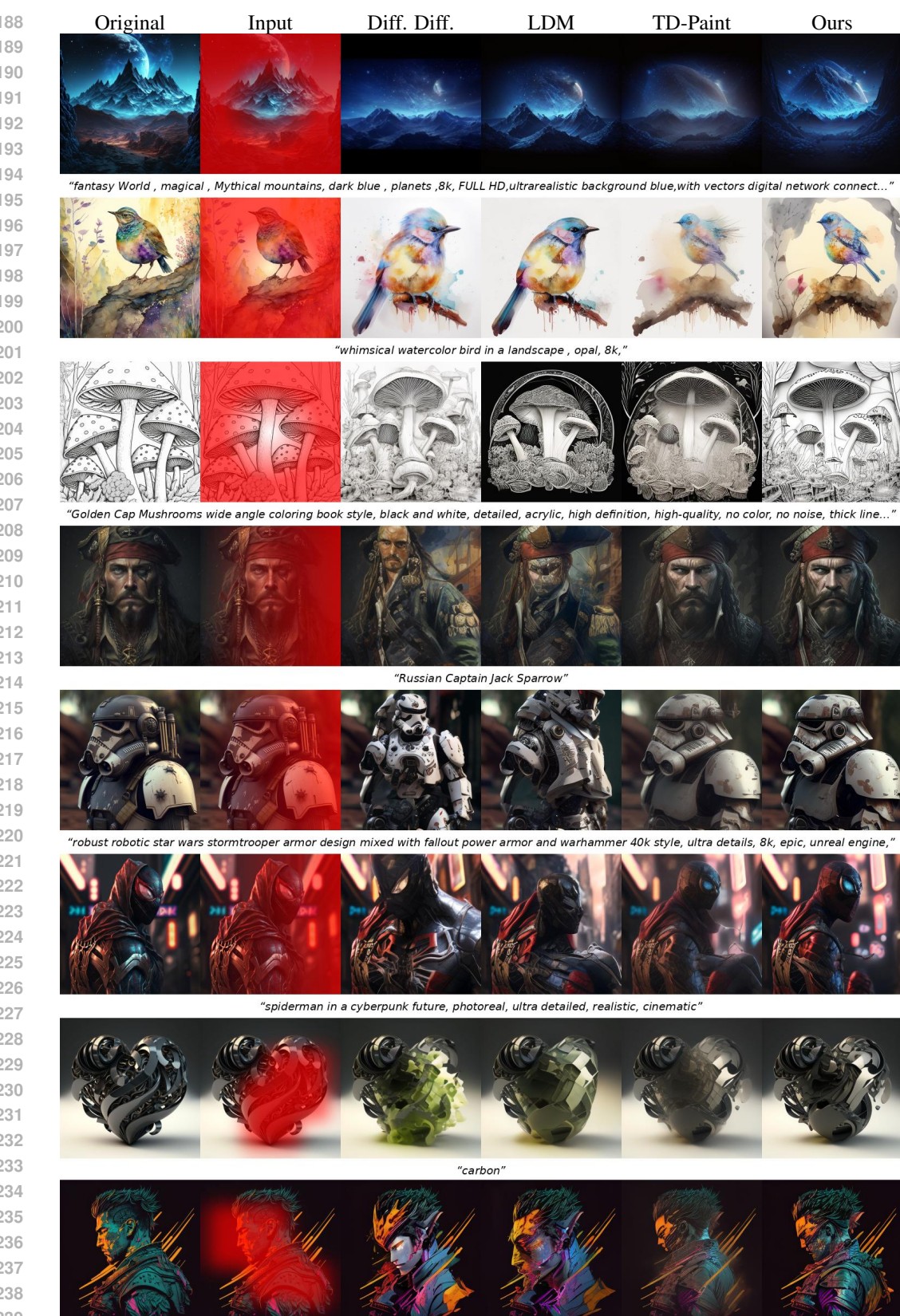

*"fantasy World , magical , Mythical mountains, dark blue , planets ,8k, FULL HD,ultrarealistic background blue,with vectors digital network connect…"*

*"whimsical watercolor bird in a landscape , opal, 8k,"*

*"Golden Cap Mushrooms wide angle coloring book style, black and white, detailed, acrylic, high definition, high-quality, no color, no noise, thick line…"*

*"Russian Captain Jack Sparrow"*

*"robust robotic star wars stormtrooper armor design mixed with fallout power armor and warhammer 40k style, ultra details, 8k, epic, unreal engine,"*

*"spiderman in a cyberpunk future, photoreal, ultra detailed, realistic, cinematic"*

*"carbon"*

*"illustration, comic, graphic, vector, pops of color, cyberpunk, samurai, profile, portrait, crop above torso, hypercolor, detailed, rimlight, high quality"*

Figure D6: **Text-Conditional Inpainting on JourneyDB with Continuous Masks.**

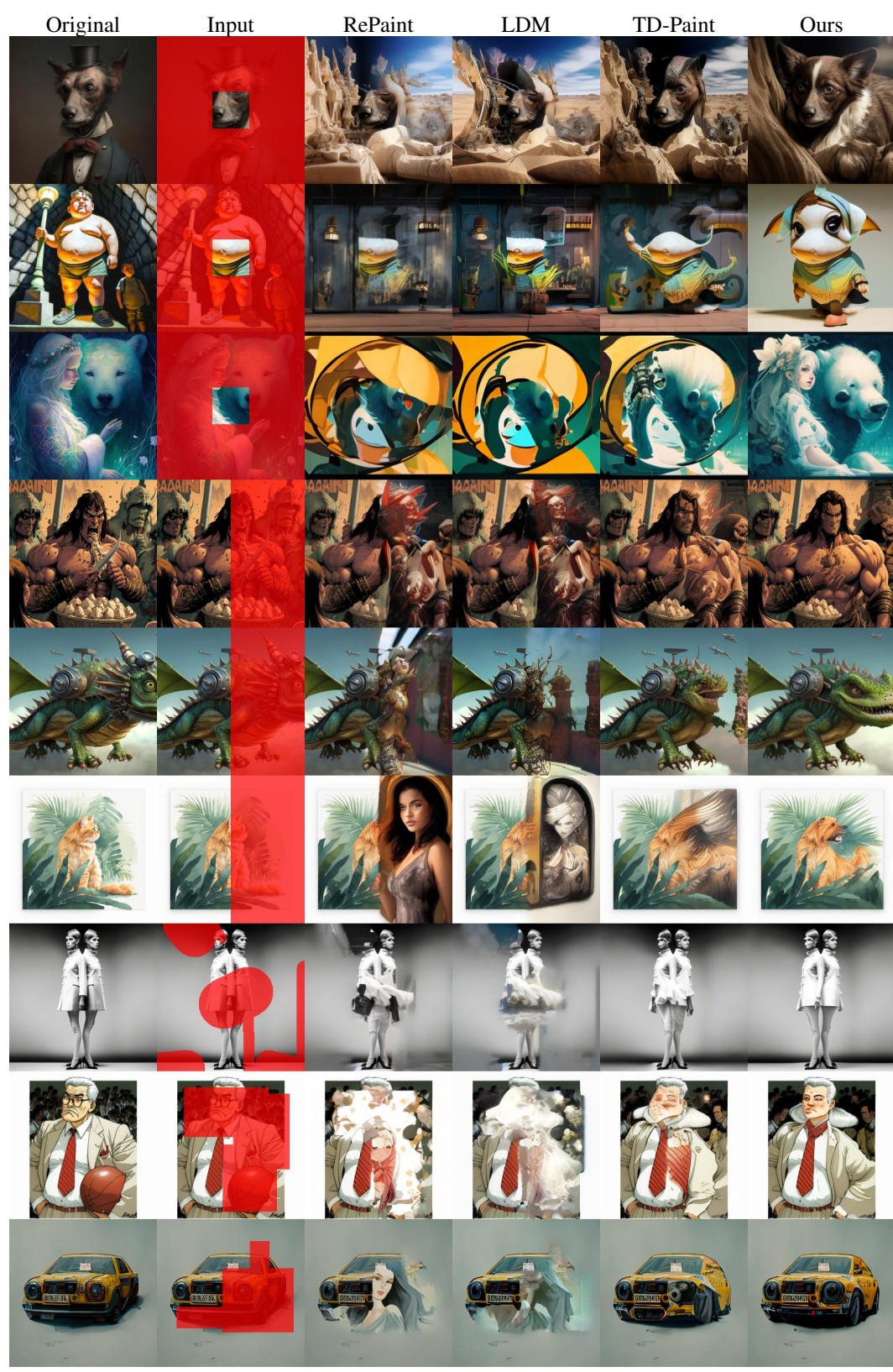

Figure D7: **Unconditional Inpainting on JourneyDB with Binary Masks.**

| Original | Input | Diff. Diff. | LDM | TD-Paint | Ours |
|----------|-------|-------------|-----|----------|------|

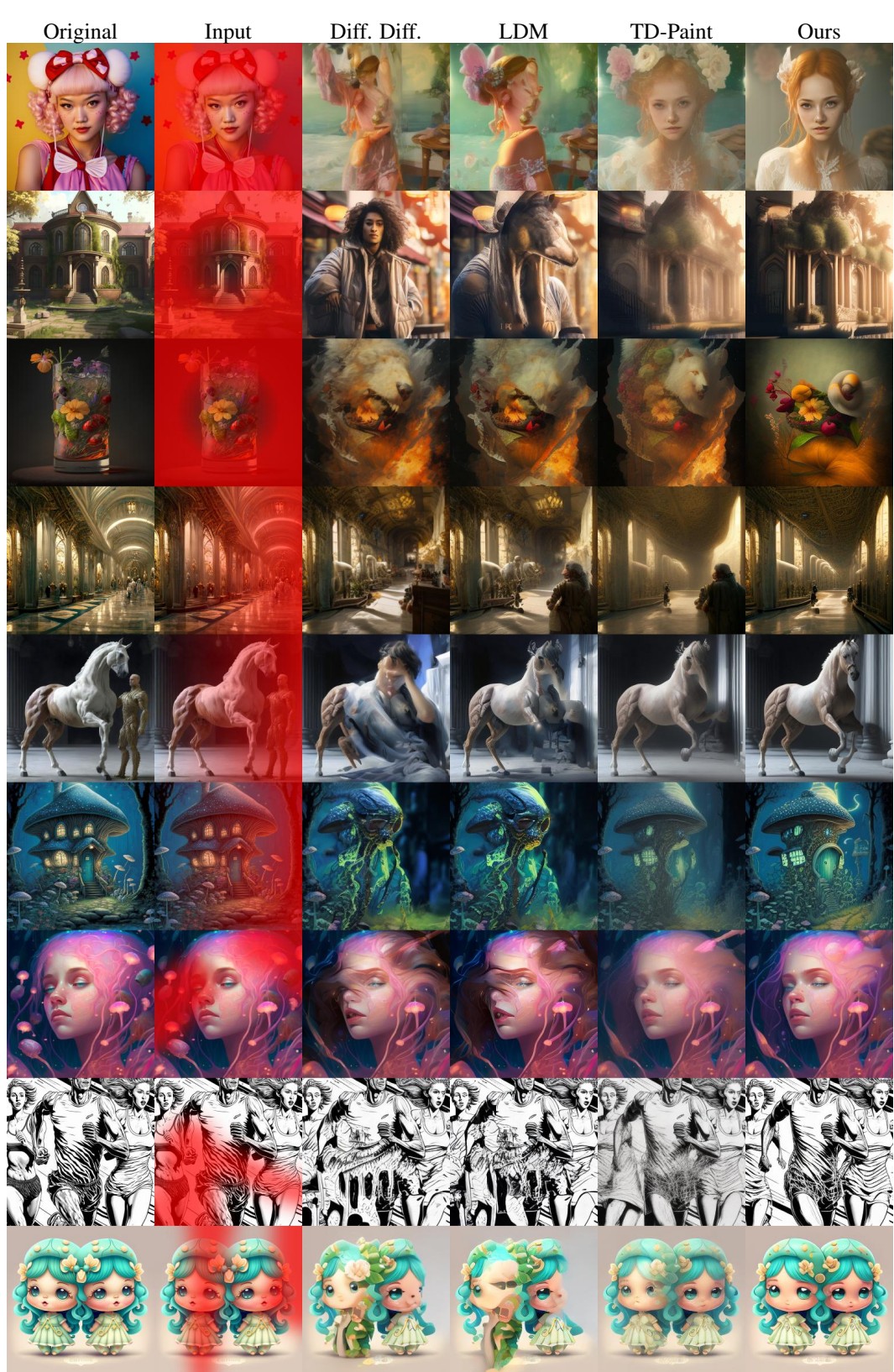

Figure D8: **Unconditional Inpainting on JourneyDB with Continuous Masks.**

