# OpenReview forum: "Saint: Spatial Guidance for Inpainting"
_ICLR.cc/2026/Conference — ICLR 2026 Conference Withdrawn Submission_

### Official Review · Reviewer_P4Wz · 2025-10-30

**Soundness:** 3
**Presentation:** 3
**Contribution:** 3
**Rating:** 2
**Confidence:** 3

**Summary:**

The paper proposes Saint, a framework for highly flexible image inpainting that leverages diffusion (or flow) transformer models. The framewor supports inpainting for arbitrary mask shapes and extends this capability to continuous-valued masks. The main methodological idea is to transform the input mask into a patch-wise denoising timestep signal. The diffusion model is then specifically fine-tuned to handle these differently noised patches.

For improved performance, the authors apply spatial classifier-free guidance (CFG), and they introduce a token caching mechanism to enable faster inference. The authors conduct a thorough experimental evaluation, comparing Saint against baselines such as RePaint, LDM, TD-Paint, and Differential Diffusion on various inpainting tasks across multiple datasets. Saint demonstrates strong performance across the board, and the beneficial effects of both CFG and the proposed caching mechanism are experimentally demonstrated.

**Strengths:**

The paper is well-written and is easy to follow. The experiments are quite comrehensive and demonstrate different claims of the paper.

**Weaknesses:**

One weakness I can see is the lack of error bars in the metrics reported in the paper. Are these results significantly different? Besides that, I have quite a few questions that are detailed in the next section.

**Questions:**

**The task definition and transformation to latent space**

- In the inpainting tasks in this paper, it is assumed that both the full image $I$ and mask $M$ are given. Then the image is encoded to $X$ by the VAE encoder and the mask is bilinearly downsampled to $T$. Therefore, the task is not to inpaint $X$ given the continuous mask $T$. However, typically in inpainting tasks only the non-masked part of the image is known. How do the authors deal with those cases, since a partial image will be out of distribution for the VAE encoder?
- A more important concern along the same lines is whether there is information leak from the non-observed parts of the image to the latent parts?

**Experiments**

While the results are quite impressive, they raise a few questions:

- Why does LDM do much worse compared to Saint on binary mask inpainting tasks? As far as I understand, at inference time, with binary masks, LDM and Saint are effectively identical. They both start with the maximum noise level for all the patches, and follow the same sampling loop. LDM just has an extra channel for the mask, which makes the model even larger than the Saint model. They are also both fine-tuned for the same number of iterations. Therefore, my expectation was that LDM does similar to Saint (if not better) on the binary masks tasks. What is the authors' comments on why it is doing much worse?

- Do the Saint model results in Table 1 and Table 2 use spatial CFG? If so, I think another row for "Saint w/o spatial CFG" should be added for a clearer picture and more fair comparison.

- Is there any reported metrics measuring the diversity of the samples and overfitting of the model? A model that simply memorizes the dataset can achieve optimal scores on the metrics in Table 1 and 2.

**Token Caching**

- The authors assume that the tokens do not change significantly between consecutive sampling steps. While the authors show the effectiveness of their method empirically, it is not clear to me why this assumption should be true, and whether this assumption is generalizable to other architectures. Do the authors have any comments on this?

I am keen to see authors' comments and willing to revise my score depending on the authors' response to my questions.

---

### Official Review · Reviewer_Z58A · 2025-10-31

**Soundness:** 3
**Presentation:** 2
**Contribution:** 2
**Rating:** 4
**Confidence:** 3

**Summary:**

This paper presents Saint, which finetunes a latent large-scale DiTs for image inpainting based on Spatial Reasoning Model. It proposes to encode binary/continuous and directly use it as the noise level instead of take it as the input to the denoising network via concatenation. It further proposes spatial classifier-free guidance, which enforces consistency between the generated samples and the history of already generated tokens. To accelerate sampling during inference time, clean token caching is introduced which only denoises tokens that needs to be denoised and save key and query for the conditioning variables. It also shows promising results in the experiment section.

**Strengths:**

I am not an expert in the inpainting field, and here are my opinons:

1. This paper apply classifier-free guidance on the top of Spatial reasoning models, which was originally used for small-scale toy problems and it is an reasonable combination to do the latent image inpainting tasks.
2. The clean token caching strategy which caches keys and values for conditioning variables speed up the sampling.
3. The experimental results show that the proposed method performs other existing work most of time.

**Weaknesses:**

1. I found it hard to parse Algorithm 1 to construct unconditional input X', until section 3.2 it verbally explains that every clean token is replaced by gaussian noise. I can see that the current algorithm looks concise, but it is difficult to understand.
2. Since we know the trade-off between diversity and fidelity using classifier-free guidance, which is also mentioned in the paper, but diversity is not measure in the paper.

**Questions:**

1. In section 3.1.2, it says 10% is used to train the denoiser for the conditional case. How is this proportion decided?
2. I am wondering how the mask size could affect the performance in the experiment. It could have an ablation study here.
3. This is an open question, but I consider this masked diffusion to inpainting could be related to discrete diffusion in LLM. What do you think about the relationship between them?
4. This paper chose to fine-tune pre-trained DiTs to avoid training from scratch, but during inference time, it requires two forward passes at each sampling step. Then the inference time is doubled and I am not sure whether the clean token caching will rescue here effectively.

---

### Official Review · Reviewer_WWAz · 2025-10-31

**Soundness:** 3
**Presentation:** 2
**Contribution:** 2
**Rating:** 4
**Confidence:** 4

**Summary:**

This paper proposes an image inpainting framework Saint based on latent multi-variable architecture, combining large-scale diffusion models with flow-based transformers. Saint fine-tunes pre-trained DiTs as Spatial Reasoning Models (SRMs) by applying different noise levels to masked and unmasked regions, enabling the model to directly condition on partially noised latent variables. This improves inpainting performance under both binary masks and continuous masks. Additionally, the proposed Spatial Classifier-Free Guidance and Clean Token Caching (CTC) strategies enhance the quality and consistency of inpainted images while accelerating fine-grained inpainting.

**Strengths:**

1. The overall writing and organization of the paper are clear and easy to follow. The proposed method is effectively presented and well-illustrated through Figures 2, 3.
2. The authors conducted sufficient experiments and outperformed SOTA methods, such as TD-Paint.
3. The proposed Clean Token Caching is interesting and significantly reduces inference time for small-scale editing scenarios.

**Weaknesses:**

1. TD-Paint also applies different noise levels to different pixels, so Saint's unique contribution in this aspect is not particularly prominent.
2. The authors did not explain why SRM can improve inpainting performance by combining masked images with input images. Therefore, the motivation for introducing SRM is not sufficiently justified.
3. The authors proposed the CTC strategy to accelerate the inpainting process, but did not compare inference efficiency with the latest methods under different patch sizes, so the contribution of CTC remains insignificant. Additionally, authors are encouraged to evaluate the memory consumption of CTC. It would better demonstrate its contribution if CTC could be validated on other inpainting frameworks.
4. It is suggested that the authors add experiments on portrait datasets, such as CelebA-HQ.
5. It is suggested that the authors add metrics like CLIP-I to evaluate consistency with the input image.

**Questions:**

See the Weakness.

---

### Official Review · Reviewer_wno3 · 2025-11-01

**Soundness:** 2
**Presentation:** 2
**Contribution:** 2
**Rating:** 4
**Confidence:** 5

**Summary:**

This paper proposes a new inpainting method based on LDM. The main motivation is to design a spatially asynchronous generation framework based on LDM. The proposed method can be viewed as an LDM counterpart of TD-Paint and RAD, which are based on pixel-space diffusion models. The paper achieves this by rearranging SRM in the spatial domain. Some interesting amendments are added, i.e., spatial "classifier-free guidance" to expedite consistency and clean token caching for reducing computational burden. Experiments demonstrate that the proposed method achieves state-of-the-art results.

**Strengths:**

- The design of the proposed method is plausible and convincing.

- State-of-the-art results.

**Weaknesses:**

- The level of novelty is somewhat low. The main part of the method relies largely on SRM and the spatially asynchronous generation idea of TD-Paint and RAD. Accordingly, the core part is somewhat straightforward.

- The spatial CFG idea is somewhat interesting, though. It is clever, and the improvement due to this is convincing. At the same time, however, it is also somewhat controversial. Is it really CFG? Indeed, time steps (especially the spatially variant ones shown in this setting) can be viewed as an indirect form of conditioning, but they are intertwined with the diffusion process and are not conditions in a classical sense. What I'm pointing out here is that, even though I'm convinced by the effectiveness, it feels like more theoretical justification is needed here. For example, the original CFG paper provides some probabilistic justification for the technique. Is the justification still valid when the conditions are replaced with time steps? Or, are there any other justifications (other than simply mimicking the technique)?

- Clean token caching is a nice touch, but it is more like adopting an already well-established technique.

**Questions:**

Please see the above weaknesses.

---

### Note · Authors · 2025-11-13

I have read and agree with the venue's withdrawal policy on behalf of myself and my co-authors.